# Changes to social feeding behaviors are not sufficient for fitness gains of the *Caenorhabditis elegans* N2 reference strain

Yuehui Zhao[1], Lijiang Long[1], Wen Xu[1], Richard F Campbell[1], Edward E Large[1], Joshua S Greene[2], Patrick T McGrath[1,3,4]*

[1]Department of Biological Sciences, Georgia Institute of Technology, Atlanta, United States; [2]The Rockefeller University, New York, United States; [3]Department of Physics, Georgia Institute of Technology, Atlanta, United States; [4]Institute of Bioengineering and Bioscience, Georgia Institute of Technology, Atlanta, United States

**Abstract** The standard reference *Caenorhabditis elegans* strain, N2, has evolved marked behavioral changes in social feeding behavior since its isolation from the wild. We show that the causal, laboratory-derived mutations in two genes, npr-1 and glb-5, confer large fitness advantages in standard laboratory conditions. Using environmental manipulations that suppress social/solitary behavior differences, we show the fitness advantages of the derived alleles remained unchanged, suggesting selection on these alleles acted through pleiotropic traits. Transcriptomics, developmental timing, and food consumption assays showed that N2 animals mature faster, produce more sperm, and consume more food than a strain containing ancestral alleles of these genes regardless of behavioral strategies. Our data suggest that the pleiotropic effects of glb-5 and npr-1 are a consequence of changes to $O_2$-sensing neurons that regulate both aerotaxis and energy homeostasis. Our results demonstrate how pleiotropy can lead to profound behavioral changes in a popular laboratory model.
DOI: https://doi.org/10.7554/eLife.38675.001

*For correspondence:
patrick.mcgrath@biology.gatech.edu

**Competing interests:** The authors declare that no competing interests exist.

## Introduction

It is tempting to compare the endless forms of life and create adaptive hypotheses to explain their differences. Why are polar bears white? Perhaps as camouflage for when they hunt. Or perhaps to make it easier to absorb heat from the sun. Both explanations make sense, but designing experiments to distinguish between these possibilities is not trivial. Further, as Gould and Lewontin critiqued, relying on adaptive evolution as the sole explanation for phenotypic change while ignoring alternative explanations such as genetic drift, adaptive constraints, or pleiotropy does not follow Darwin's pluralistic approach (*Gould and Lewontin, 1979*). In the current era, inexpensive next-generation sequencing and increasingly sophisticated bioinformatics analysis enable the identification of causative mutations with signatures of selection, yet it is difficult to determine *why* these alleles are under selection. Indeed, the pervasive effects of pleiotropy means that signatures of selection alone are not enough, adaptive hypotheses must be tested directly. Experimental evolution offers a route to test hypothesis directly (*Fisher and Lang, 2016*; *Lenski, 2017*; *Teotónio et al., 2017*). The ability to manipulate model organisms in the lab provides a greater opportunity to test adaptive hypotheses beyond arguments of plausibility and address the role of these other competing themes in the evolution of biological traits.

**eLife digest** Why do humans walk on two feet? And what makes us smarter than our ape ancestors? The answers to these questions, and countless others about the particular traits of any number of species, is often said to be natural selection – a process where genes that ensure the survival of a species are favored of others. But it is not always the answer. Other evolutionary forces, such as random changes to the frequency of certain gene variants, restrictions on the development of a certain trait and pleiotropy (where one gene influences other, seemingly unrelated traits) can also cause differences between species.

Designing experiments to test whether a trait difference is due to natural selection or other factors is notoriously difficult. However, the humble nematode worm, *Caenorhabditis elegans*, has proven to be particularly useful in this respect. One subtype or strain of *C. elegans* with certain changes to its genes is used internationally as a 'reference strain', to ensure results between labs are comparable. This strain, N2, has been bred in the laboratory for hundreds of generations, isolated from its wild counterparts. N2 shows several differences in behavior from the wildtype, including its feeding habits. Wild *C. elegans* tend to feed together socially, whereas N2 prefers to feed alone.

In 1998 and 2009, researchers – including some involved in the current study – have identified the genetic modifications responsible for this change in behavior. Now, Zhao et al. set out to determine whether this was due to natural selection, and if so, was there a benefit to solitary feeding in laboratory conditions that was driving this genetic change? Zhao et al. found that the genetic changes in the N2 strain gave the worms a considerable advantage in the artificial environment. However, experiments to modify the conditions the animals grew in revealed that the solitary feeding habits were not necessary for the fitness advantage. In other words, the changes in feeding habits were a symptom of the genetic changes that gave N2 a selective advantage, but they were not the cause. In other words, the changes in feeding behavior were not a result of natural selection, but rather of pleiotropy.

The findings highlight that not every change in a trait is down to natural selection and must therefore be put to the test. With declining costs of DNA sequencing, researchers can now easily identify genes and regions of DNA that are likely to be under selection. However, they must be careful before leaping to the conclusion that behavioral differences linked to genetic changes are adaptive. In addition, the findings show that the laboratories relying on N2 as a model organism should be aware that the strain has evolved fundamental differences in its brain connections compared with the wildtype.

DOI: https://doi.org/10.7554/eLife.38675.002

These studies are also useful for understanding how organisms adapt to laboratory conditions. Since the fundamental work of Gregor Mendel elucidating the laws of genetic transmission, model organisms have enabled experimenters to gain fundamental insights into many biological processes. Modern research tools are facilitating the use of new and unusual species to analyze longstanding biological questions (*Alfred and Baldwin, 2015*; *Gladfelter, 2015*; *Goldstein and King, 2016*; *Russell et al., 2017*). More and more species are reared in the laboratory as models for biological traits of interest. An issue for these approaches, particularly for comparative analysis or for those addressing evolutionary questions, is the extreme shift in environment and associated selective pressures that these populations experience. All species evolve through the process of natural selection and genetic drift; many model organisms have evolved by exposure to the novel and artificial conditions experienced in the lab (*Orozco-terWengel et al., 2012*; *Duveau and Félix, 2012*; *Goto et al., 2013*; *Kasahara et al., 2010*; *Marks et al., 2010*; *Stanley and Kulathinal, 2016*; *Yvert et al., 2003*). Understanding the process of adaptation of wild populations to captivity is necessary to understand how the genetic, developmental, and neural circuits are changed in these laboratory populations.

As a model for understanding laboratory adaptation in a multicellular organism, we have focused our studies on the N2 strain of *Caenorhabditis elegans*. N2 is the canonical reference strain used by hundreds of *C. elegans* labs across the world. While this strain was introduced to the genetics research community by Sydney Brenner in 1974 (*Brenner, 1974*), it was actually isolated by L.N.

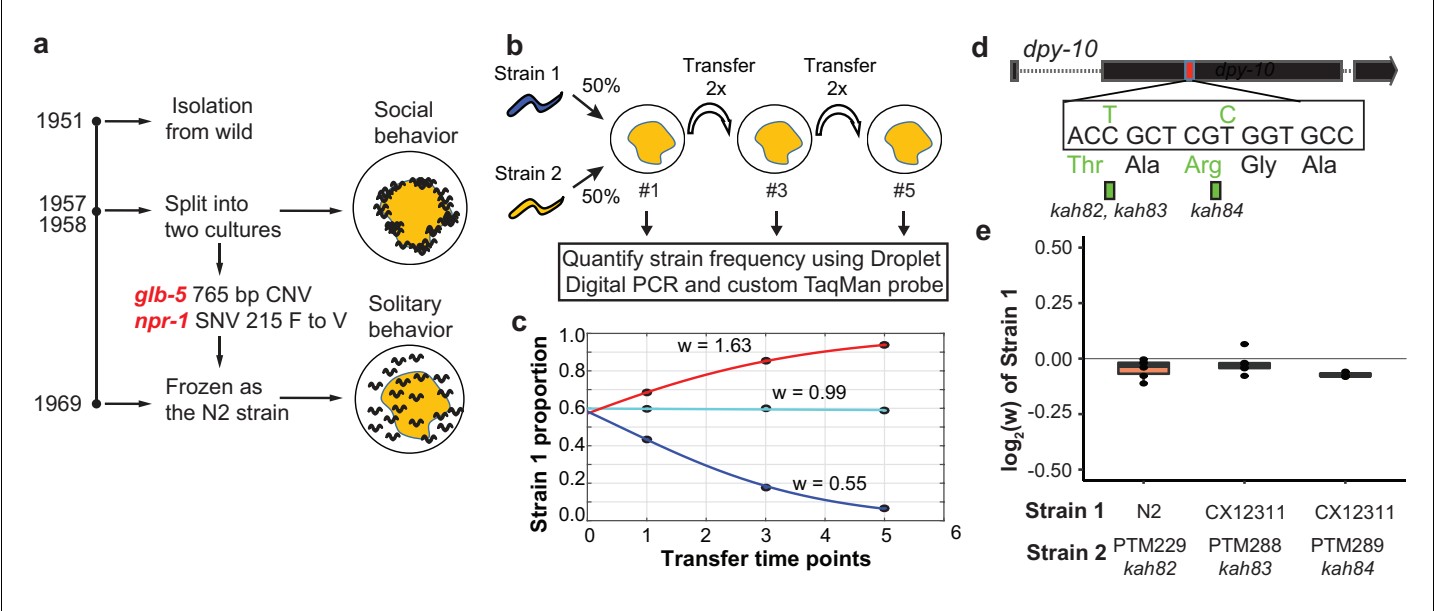

**Figure 1.** Schematic of competition assays used to measure relative fitness levels between two strains. (**a**) Overview of life history of the standard reference N2 strain since its isolation from the wild. Derived alleles in *npr-1* and *glb-5* arose and fixed after 1957 and before 1969 when methods for cryopreservation were developed. These two alleles were identified for their role in changing foraging behavior on bacterial lawns from social to solitary behavior. (**b**) Schematic of pairwise competition experiments used throughout the paper to quantify fitness differences between two strains. (**c**) Relative proportion of each strain as ascertained by Droplet Digital PCR using a custom TaqMan probe (dots) is used to estimate the relative fitness between the two strains (line). (**d**) Silent mutations were edited into the 90th or 92nd amino acid of the *dpy-10* gene using CRISPR/Cas9 to create a common SNV for Droplet Digital PCR. We refer to these as barcoded strains. (**e**) Competition experiments between the parent strain (top) and the same strain containing one of the silent mutations. We display the result from each competition experiment as a single dot overlaid on top of a boxplot showing the mean, first, and third quartiles of all replicates.

DOI: https://doi.org/10.7554/eLife.38675.003

The following source data is available for figure 1:

**Source data 1.** Relative proportion of each strain as ascertained by Droplet Digital PCR shown in *Figure 1c*.
DOI: https://doi.org/10.7554/eLife.38675.004

**Source data 2.** Competition experiment using indicated barcoded strains carry the *dpy-10* silent mutation shown in *Figure 1e*.
DOI: https://doi.org/10.7554/eLife.38675.005

Staniland and Warwick Nicholas from mushroom compost in 1951, spending multiple decades (~300–2000 generations) in two primary growth conditions: on agar plates where bacteria was its primary food source or in liquid axenic media (*Sterken et al., 2015*). A small number (~100) of new mutations that arose and fixed in the N2 strain following isolation from the wild have been identified (*McGrath et al., 2011*), including a neomorphic, missense mutation in the neuropeptide receptor gene *npr-1* and a recessive, 765 bp duplication in the nematode-specific globin gene *glb-5*. These mutations were originally identified for their role in foraging and aerotaxis behaviors and were initially thought to represent natural genetic variants (*de Bono and Bargmann, 1998*; *Persson et al., 2009*) (*Figure 1a*). A large body of work has found that these genes regulate the activity of the URX-RMG neuronal circuit that controls $O_2$ responses on food (*Chang et al., 2006*; *Coates and de Bono, 2002*; *Gray et al., 2004*; *Macosko et al., 2009*; *McGrath et al., 2009*; *Persson et al., 2009*). Animals with the ancestral alleles of *npr-1* and *glb-5* prefer ~10% $O_2$ concentrations while foraging and follow $O_2$ gradients to the border of bacterial lawns (~12% $O_2$) and feed in groups (called social behavior); animals containing the derived alleles of these genes are less sensitive to 10–21% $O_2$ gradients in the presence of food and feed alone (called solitary behavior) (*Chang and Bargmann, 2008*; *Chang et al., 2006*; *Gray et al., 2004*).

We have previously proposed that the derived alleles of *glb-5* and *npr-1* were fixed by selection as solitary animals are more likely to be picked when propagating animals to new plates (*McGrath et al., 2009*). However, aggregation behavior in the ancestral *npr-1* strain appears to

create local food depletion leading to a weak starvation state, which reduces reproduction and growth (*Andersen et al., 2014*). Potentially, this starvation difference could be responsible for the fitness differences of the strains. Consistent with both hypotheses, a number of experimental crosses or competition experiments between parental strains that are polymorphic for *npr-1* have resulted in enrichment of the derived allele of *npr-1*, suggesting it confers a fitness advantage under standard lab husbandry (*Gloria-Soria and Azevedo, 2008*; *Noble et al., 2017*; *Weber et al., 2010*).

In order to distinguish between these hypotheses, we performed pairwise competition experiments following a number of environmental and/or genetic manipulations. Surprisingly, our results suggest that neither hypothesis is correct. While the derived *npr-1* and *glb-5* alleles increase fitness of animals on agar plates, the differences in social vs. solitary behavior are not necessary for their differences in fitness. Instead, our work suggests that fitness gains are due to increases in food consumption and changes in reproductive timing, mediated by $O_2$-sensing body cavity neurons that are also required for social feeding behaviors. Our work demonstrates that even when alleles are identified that confer fitness advantages, care must be taken in inferring the phenotypes that are responsible due to the pleiotropic actions of genetic changes.

## Results

### Derived alleles of *npr-1* and *glb-5* increase fitness in laboratory conditions

In previous reports, we have used multigenerational pairwise competition experiments to compare the relative fitness of two strains (*Figure 1b*) utilizing Droplet Digital PCR with a custom TaqMan probe to quantify the proportion of each genotype (*Evans et al., 2017*; *Greene et al., 2016*; *Large et al., 2016*). To quantify this change, we used a generic selection model to estimate the relative fitness difference (w) between the two strains (*Figure 1c*). In this context, relative fitness measures the generational change in relative abundance of each of the two strains. We also used CRISPR-enabled genome engineering to create strains with a silent mutation in the *dpy-10* gene using a previously published high-efficiency guide RNA (*Figure 1d*) (*Arribere et al., 2014*), which we will refer to as barcoded strains. These strains allow us to use a common Taqman probe to quantify the relative fitness of a test strain against these barcoded strains. We confirmed that the *dpy-10* silent mutation had no statistically significant effect on fitness in two genetic backgrounds studied throughout this report (*Figure 1e*).

In order to test the fitness effect of the derived alleles of *npr-1* and *glb-5*, we utilized three previously described near isogenic lines (NILs) containing ancestral alleles of *npr-1* (QG1), *glb-5* (CX10774), or both genes (CX12311) introgressed from the Hawaiian CB4856 wild strain into the standard N2 background (*Bernstein and Rockman, 2016*; *McGrath et al., 2009*; *McGrath et al., 2011*). The *npr-1* introgressed region is ~110 kb in size and the *glb-5* introgressed region is ~290 kb in size. For brevity, we will refer to genotype of these introgressed regions throughout this report by the ancestral/derived allele they contain (e.g. the ancestral allele of *npr-1* vs the introgressed region containing the ancestral allele of *npr-1*). For clarity, we will refer to the NILs colloquially using the ancestral introgression(s) they contain instead of their opaque strain names (i.e. N2 = N2, CX10774 = N2$_{glb-5}$, QG1 = N2$_{npr-1}$, and CX12311 = N2$_{glb-5, npr-1}$). If needed, readers can find the strain name used in each figure in the supplemental source data files. In contrast to the N2 strain, the N2$_{glb-5, npr-1}$ strain aggregates at the border of bacterial lawns where $O_2$ levels are lowest due to the increased height of the bacterial lawn. We confirmed previous reports that both the derived alleles of *npr-1* and *glb-5* suppress bordering behavior to varying degrees (*Bendesky et al., 2012*; *de Bono and Bargmann, 1998*; *McGrath et al., 2009*); *npr-1* accounted for the majority of the difference with *glb-5* playing a modulatory role (*Figure 2a*). To compare the relative fitness of the four strains, we competed each strain against the barcoded N2$_{glb-5, npr-1}$ strain, transferring animals each generation by washing to minimize potential sources of investigator bias toward picking social or solitary animals (*Figure 2b*). The N2 strain was the most fit in these conditions, with a relative fitness (w) of ~1.30. Interestingly, the fitness effects of the *glb-5* and *npr-1* regions were epistatic - the derived allele of *glb-5* increased the relative fitness in the derived *npr-1* background but showed no effect in the ancestral allele of *npr-1*. The derived *npr-1* allele increased fitness in both backgrounds of *glb-5*. To confirm the fitness advantage of the derived *glb-5* allele in the derived *npr-1*

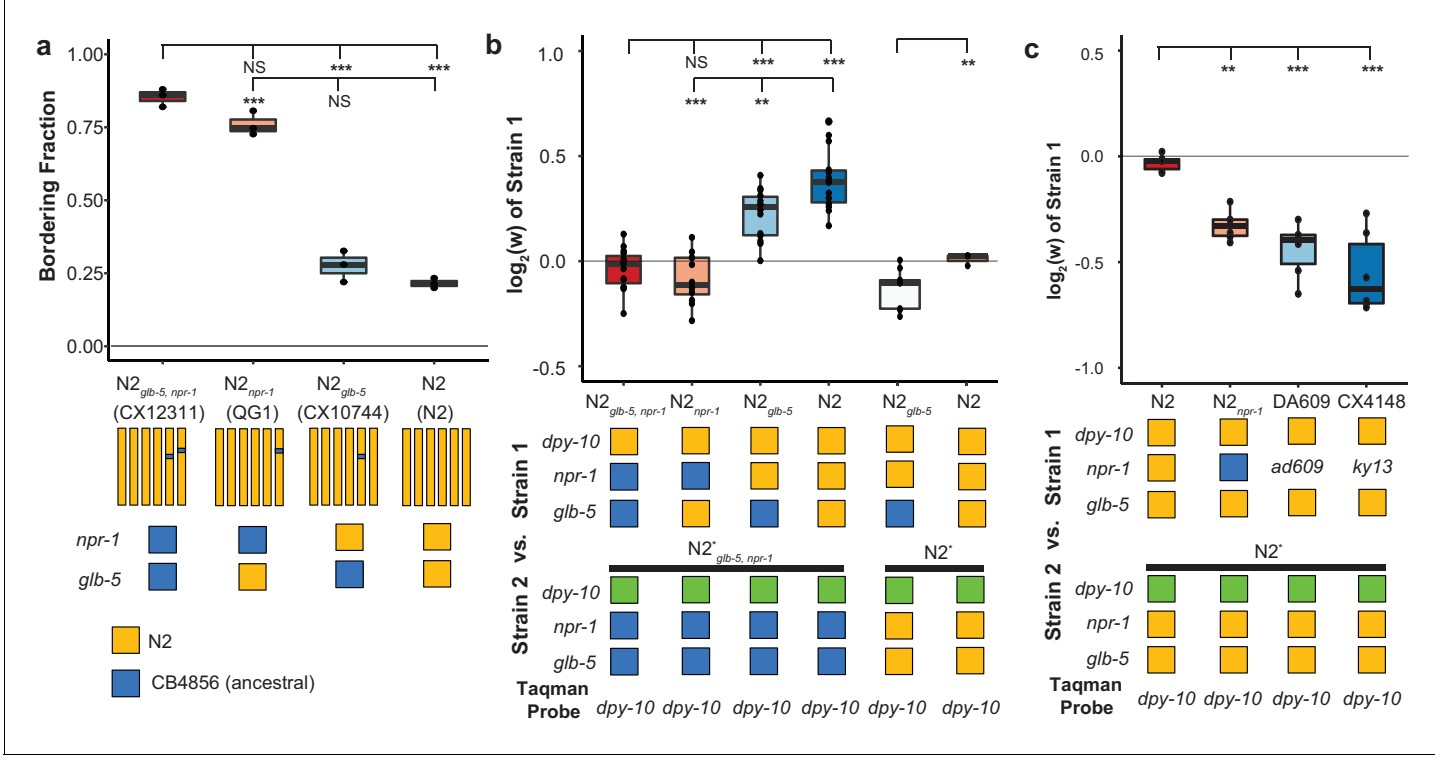

**Figure 2.** Derived alleles of *npr-1* and *glb-5* are beneficial. (**a**) The bordering rate of the N2 reference strain compared to three near isogenic lines (NILs) containing ancestral alleles of *npr-1* and/or *glb-5* introgressed from the CB4856 wild strain. Bordering rate is defined as the fraction of animals on the edge of the bacterial lawn at a single timepoint. Schematic of each NIL shown below along with the allele of *npr-1* and *glb-5* they contain. Orange represents N2-derived DNA and blue represents CB4856-derived DNA. These strains are referred to by the ancestral alleles they contain (e.g. N2$_{glb-5}$=CX10744, which is an introgression surrounding *glb-5*). To ascertain statistical significance, ANOVA was used followed by a Tukey's Honest Significant Difference test for multiple comparison tests. NS, not significant, **p<0.01, ***p<*0.001*. (**b**) Competition experiments between NILs shown in panel a against barcoded strains shown in *Figure 1d,e*. Green box indicates the strain contains the barcoded allele of *dpy-10*. Positive values indicate Strain one is more fit; negative values indicate Strain two is more fit. NS not significant, **p<0.01, ***p<*0.001* by ANOVA with Tukey's Honest Significant Difference test or Wilcoxon-Mann-Whitney nonparametric test. (**c**) Competition experiments between strains containing two loss-of-function alleles of *npr-1* (*ad609* and *ky13*) along with controls. **p<0.01, ***p<*0.001* by ANOVA with Tukey's Honest Significant Difference test.

DOI: https://doi.org/10.7554/eLife.38675.006

The following source data is available for figure 2:

**Source data 1.** The bordering rate of the N2 compared to three near isogenic lines (NILs) containing ancestral alleles of *npr-1* and/or *glb-5* introgressed from the CB4856 wild strain shown in *Figure 2a*.
DOI: https://doi.org/10.7554/eLife.38675.007

**Source data 2.** Competition experiments between N2 and NILs shown in *Figure 2b*.
DOI: https://doi.org/10.7554/eLife.38675.008

**Source data 3.** Competition experiments between strains containing two loss-of-function alleles of *npr-1* (*ad609* and *ky13*) along with N2 shown in *Figure 2c*.
DOI: https://doi.org/10.7554/eLife.38675.009

background, we also competed the N2$_{glb-5}$ strain against the barcoded N2 strain (*Figure 2b*). The estimated selective coefficient (a common measure of the fitness difference of a beneficial allele) of the *glb-5* allele in the *npr-1* derived background was s = 0.10 (0.06–0.13 95% confidence interval), the estimated selective coefficient of the *npr-1* allele in the *glb-5* ancestral background was s = 0.17 (0.12–0.23 95% confidence interval), and the estimated selective coefficient of the *npr-1* allele in the *glb-5* derived background was s = 0.30 (0.27–0.34 95% confidence interval). These selective coefficients are comparable to beneficial alleles identified in other organisms, such as the haplotype responsible for lactase persistence (~0.01–0.19) (*Bersaglieri et al., 2004*) and the sickle-cell trait (0.05–0.18) in humans (*Li, 1975*).

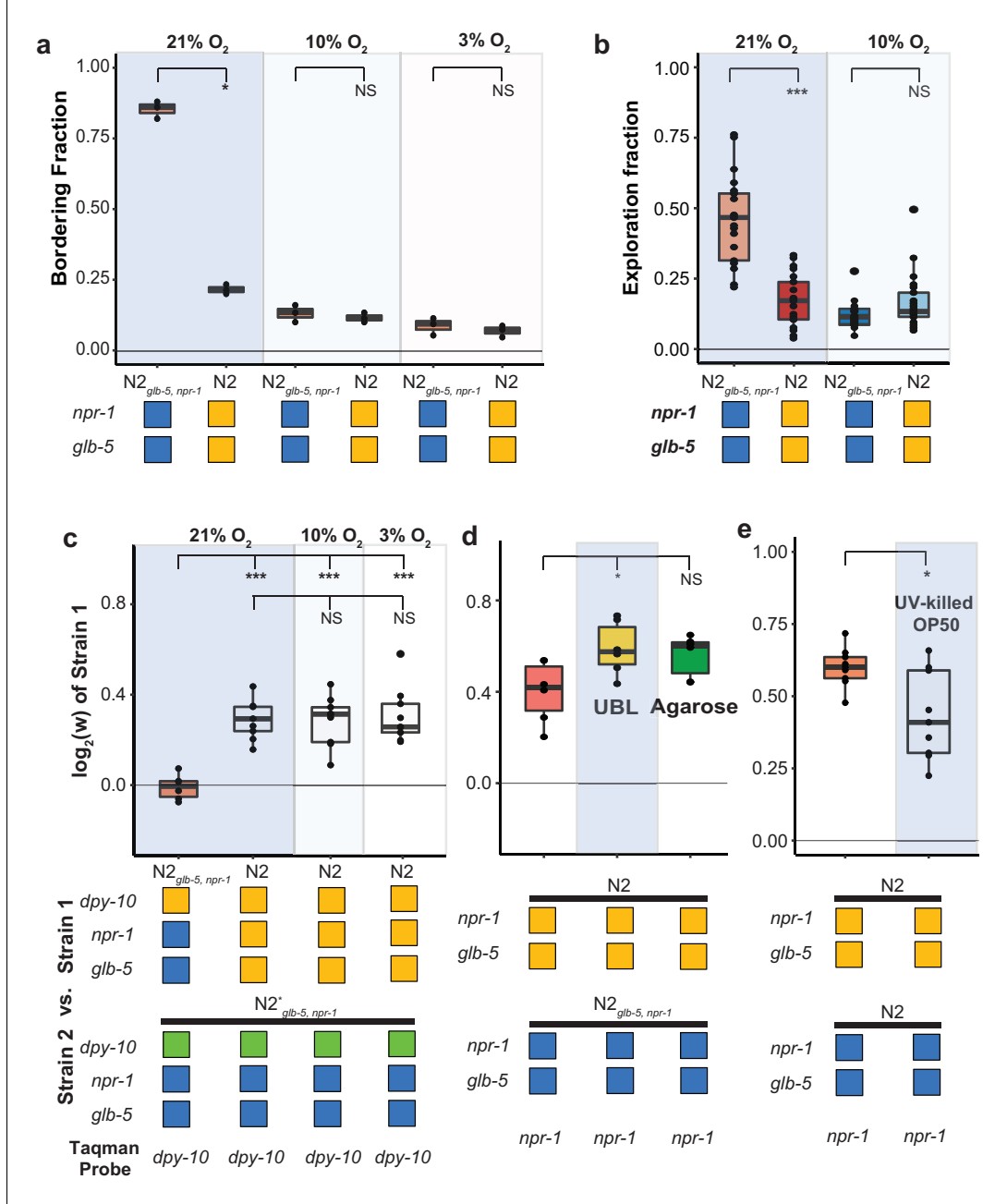

**Figure 3.** Fitness advantage of N2 is independent of foraging behavior. (**a** and **b**). Environmental $O_2$ levels were manipulated using a Biospherix chamber. Differences in (**a**) bordering behavior and (**b**) roaming and dwelling behavior were suppressed in N2$_{glb-5, npr-1}$ at lower environmental $O_2$ levels. NS not significant, *p<0.05 by Wilcoxon-Mann-Whitney nonparametric test. (**c**). Fitness advantage of N2 over the barcoded N2$_{glb-5, npr-1}$ strain was independent of environmental $O_2$. NS, not significant, ***p<0.001 by ANOVA with Tukey's Honest Significant Difference test. (**d** and **e**) Fitness differences of N2 and N2$_{glb-5, npr-1}$ on (**d**) uniform bacterial lawns (UBL) where animals were unable to border, on plates containing agarose to prevent burrowing behaviors (NS, not significant, *p<0.05 by ANOVA with Tukey's Honest Significant Difference test), and (**e**) on UV-killed bacteria (*p<0.05 by Wilcoxon-Mann-Whitney nonparametric test).

DOI: https://doi.org/10.7554/eLife.38675.010

The following source data is available for figure 3:

**Source data 1.** Bordering rate at ambient (21%) and lower environmental (10%) $O_2$ levels shown in *Figure 3a*.
DOI: https://doi.org/10.7554/eLife.38675.011
**Source data 2.** Roaming and dwelling behavioral assay in ambient (21%) and lower environmental (10%) $O_2$ levels shown in *Figure 3b*.
DOI: https://doi.org/10.7554/eLife.38675.012

*Figure 3 continued on next page*

*Figure 3 continued*

**Source data 3.** Fitness advantage of N2 over the barcoded N2$_{glb-5, npr-1}$ strain was independent of environmental O$_2$ shown in *Figure 3c*.
DOI: https://doi.org/10.7554/eLife.38675.013
**Source data 4.** Fitness differences of N2 and N2$_{glb-5, npr-1}$ on uniform bacterial lawns (UBL) and on plates containing agarose shown in *Figure 3d*.
DOI: https://doi.org/10.7554/eLife.38675.014
**Source data 5.** Fitness differences of N2 and N2$_{glb-5, npr-1}$ on UV-killed bacteria shown in *Figure 3e*.
DOI: https://doi.org/10.7554/eLife.38675.015

While the introgressions surrounding the *npr-1* and *glb-5* genes are relatively small, these NIL strains carry additional polymorphisms in surrounding genes from the CB4856 strain. We also performed competition experiments using two previously published *npr-1* loss-of-function alleles (*ad609* and *ky13*) (*de Bono and Bargmann, 1998*) against the N2 barcoded strains. Both the *npr-1(ad609)* and *npr-1(ky13)* loss-of-function alleles decreased the animal's relative fitness in an amount comparable to the ancestral allele (*Figure 2c*). We did not perform similar experiments on the *glb-5* gene. Altogether, our work suggests that the *npr-1* derived allele increases fitness of animals in laboratory conditions and also suggests that the derived allele of *glb-5* increases the fitness of animals in a *npr-1*-dependent manner.

## Suppression of social/solitary behavior differences between N2 and CX12311 does not suppress their fitness differences

Animals with reduced function of *npr-1* sense environmental O$_2$ levels and aerotax towards their preferred O$_2$ levels (10%) in the presence of foods, which results in aggregation of animals at the borders of the lawn (*Chang et al., 2006*; *Cheung et al., 2005*; *Gray et al., 2004*). This behavior can be suppressed by lowering environmental O$_2$ levels to the animals preferred O$_2$ concentrations (*Gray et al., 2004*). We decided to use this environmental manipulation to test the hypothesis that the social foraging behavior was necessary for the fitness disadvantage experienced by strains containing the ancestral alleles of *npr-1* and *glb-5*. Our above experiments hinted that this hypothesis might be incorrect as the derived *glb-5* allele reduced bordering behavior in the ancestral *npr-1* background without an associated increase in fitness. We first confirmed that we could suppress the bordering behavior differences between N2$_{glb-5, npr-1}$ and N2 by reducing environmental O$_2$ levels to 10% or 3% using a Biospherix chamber (*Figure 3a* and *Videos 1–4*). N2$_{glb-5, npr-1}$ animals did not form any social groups in the center of the lawn at the lowered O$_2$ levels and were also indistinguishable from N2 by visual inspection. We also verified that this O$_2$ manipulation also suppressed

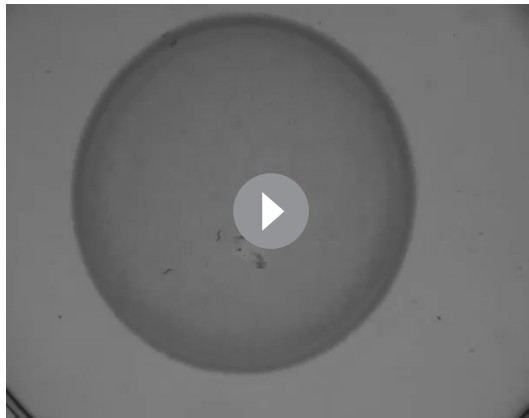

**Video 1.** N2$_{glb-5, npr-1}$ animal's behavior in 10% O$_2$ level. A single generation (3 days) of growth of the N2$_{glb-5, npr-1}$ strain in the presence of 10% environmental O$_2$.
DOI: https://doi.org/10.7554/eLife.38675.016

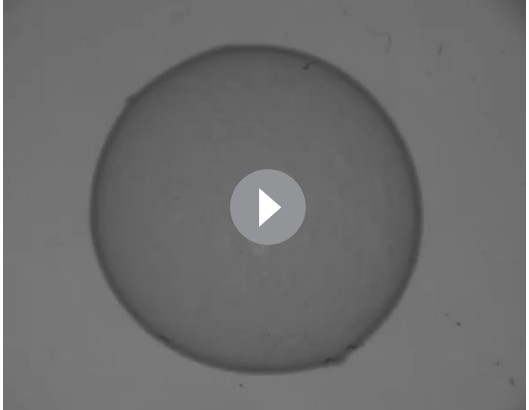

**Video 2.** N2$_{glb-5, npr-1}$ animal's behavior in 21% O$_2$ level. A single generation (3 days) of growth of the N2$_{glb-5, npr-1}$ strain in the presence of 21% environmental O$_2$.
DOI: https://doi.org/10.7554/eLife.38675.017

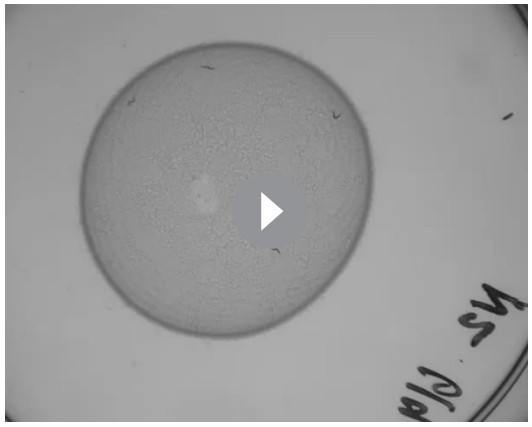 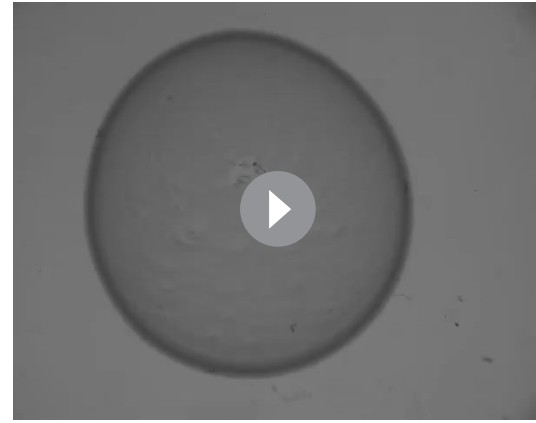

**Video 3.** N2 animal's behavior in 10% $O_2$ level. A single generation (3 days) of growth of the N2 strain in the presence of 10% environmental $O_2$.
DOI: https://doi.org/10.7554/eLife.38675.018

**Video 4.** N2 animal's behavior in 21% $O_2$ level. A single generation (3 days) of growth of the N2 strain in the presence of 21% environmental $O_2$.
DOI: https://doi.org/10.7554/eLife.38675.019

roaming/dwelling behavior (*Figure 3b*). While feeding, *C. elegans* worms alternate between bouts of active exploration (roaming) and periods of inactive movement (dwelling). Animals that are mutant for *npr-1* show increased amounts of roaming behavior (*Stern et al., 2017*).

Despite the behavioral similarity of these animals at these lower $O_2$ levels, the relative fitness differences between the N2 and N2$_{glb-5, \, npr-1}$ strains remained (*Figure 3c*). To further confirm that aggregation behavior was not necessary for the fitness differences, we also performed competition experiments on uniform bacterial lawns (UBLs), which are constructed so that the entire plate is covered with a thin bacterial lawn to remove the $O_2$ gradients created by the unequal thickness of bacteria in normal lawns. UBLs have been used to suppress *npr-1*-dependent differences in survival in response to bacterial pathogens (*Reddy et al., 2009*); however, the UBLs were unable to suppress the fitness advantage of N2 animals (*Figure 3d*).

Animals that carry the ancestral *npr-1* allele can burrow into agar when food is depleted (*de Bono and Bargmann, 1998*), raising the possibility that the fitness gains of N2 could be a result of the transfer process, which selects for animals on the surface of plates. While visual inspection of the two strains at 10% and 3% did not reveal any obvious differences in burrowing behavior, we also tested the role of burrowing in the fitness differences more rigorously by using modified nematode growth plates that contain agarose that prevents burrowing (*Andersen et al., 2014*). The relative fitness differences between N2 and N2$_{glb-5, \, npr-1}$ remained unchanged (*Figure 3d*).

Finally, we tested whether differences in resistance to infection could be responsible for the differences in fitness. The *E. coli* bacterial strain that is used to feed *C. elegans*, OP50, can also infect and kill animals, resulting in a decreased lifespan (*Garigan et al., 2002*; *Gems and Riddle, 2000*). Both *glb-5* and *npr-1* have been implicated in innate immunity and survival to pathogen exposure (*Andersen et al., 2014*; *Reddy et al., 2009*; *Styer et al., 2008*; *Zuckerman et al., 2017*). However, the fitness advantage of the N2 strain compared to the N2$_{glb-5, \, npr-1}$ strain remained when animals were competed against each other on OP50 bacteria killed by ultraviolet radiation (*Figure 3e*). The relative fitness on killed OP50 bacteria was slightly decreased; however, this could reflect differences in population demographics, as the killed OP50 supported less overall growth per plate.

These experiments motivated us to also test the relative fitness differences of 11 other wild strains isolated from different parts of the world using strains provided by the *C. elegans* Natural Diversity Resource (*Cook et al., 2017*). Each strain was competed against a barcoded N2$_{glb-5, \, npr-1}$. Consistent with their *npr-1* genotype, these wild strains all aggregated at the borders of the bacterial lawn (*Figure 4a*), but their relative fitness differences varied wildly (*Figure 4b*). The relative fitness of two of the strains (CB4856 and DL238) was greatly reduced compared to the N2$_{glb-5, \, npr-1}$ strain. The relative fitness of five of the strains were comparable to the N2. The relative fitness of the remaining four strains was statistically indistinguishable from the barcoded N2$_{glb-5, \, npr-1}$. These

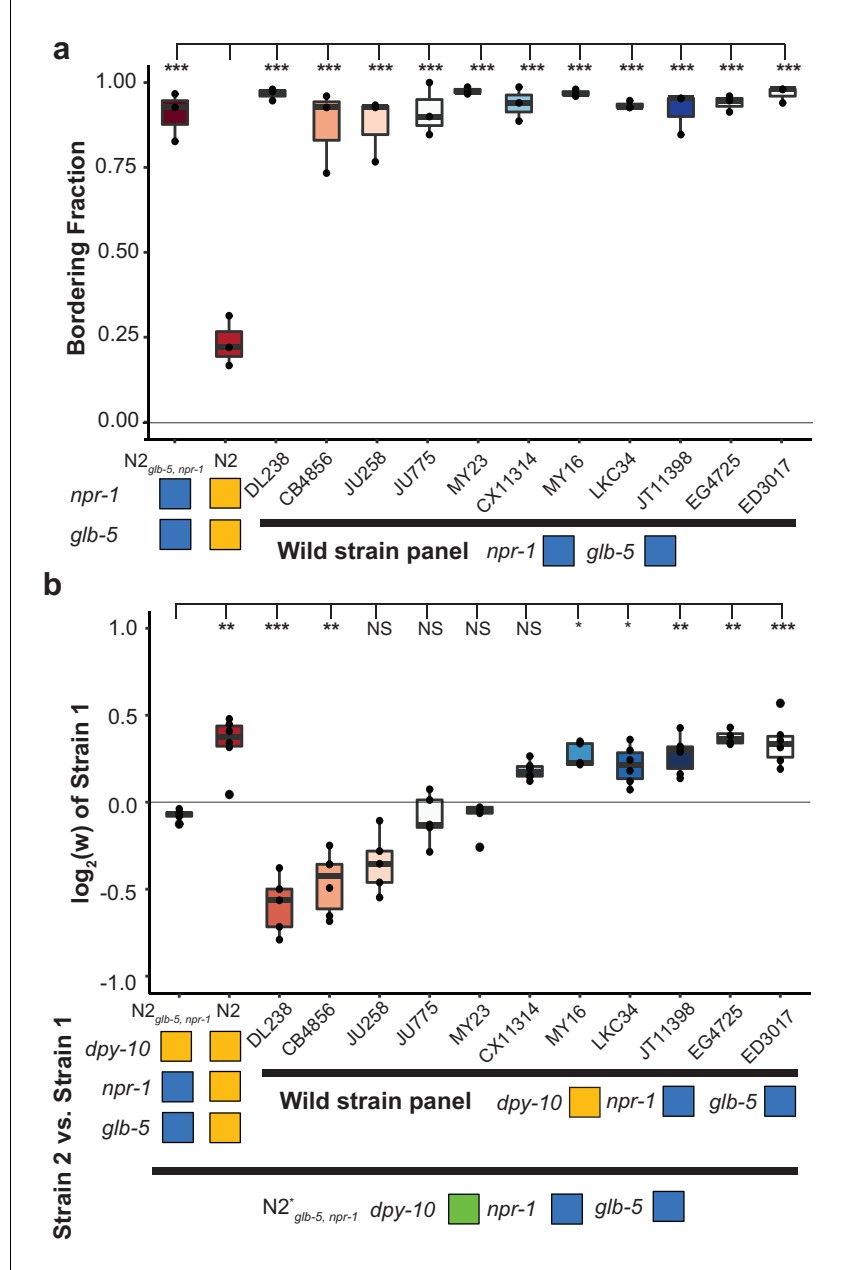

**Figure 4.** Bordering rate and relative differences between wild *C.elegans* strains. (**a**) A panel of 11 wild strains was tested for bordering behavior. Each of these wild strains contains ancestral alleles of *glb-5* and *npr-1*. ***p<0.001 by ANOVA with Tukey's Honest Significant Difference test. (**b**) Competition experiments between 11 wild strains and barcoded N2$_{glb-5,\ npr-1}$ animals. Despite the similarity of bordering behavior, these wild strains displayed a range of relative fitness. NS, not significant, *p<0.05, **p<0.01, ***p<0.001 by ANOVA with Tukey's Honest Significant Difference test.

DOI: https://doi.org/10.7554/eLife.38675.020

The following source data is available for figure 4:

**Source data 1.** Bordering rate of 11 wild strains shown in *Figure 4a*.
DOI: https://doi.org/10.7554/eLife.38675.021

**Source data 2.** Competition experiments between 11 wild strains and barcoded N2$_{glb-5,\ npr-1}$ animals shown in *Figure 4b*.
DOI: https://doi.org/10.7554/eLife.38675.022

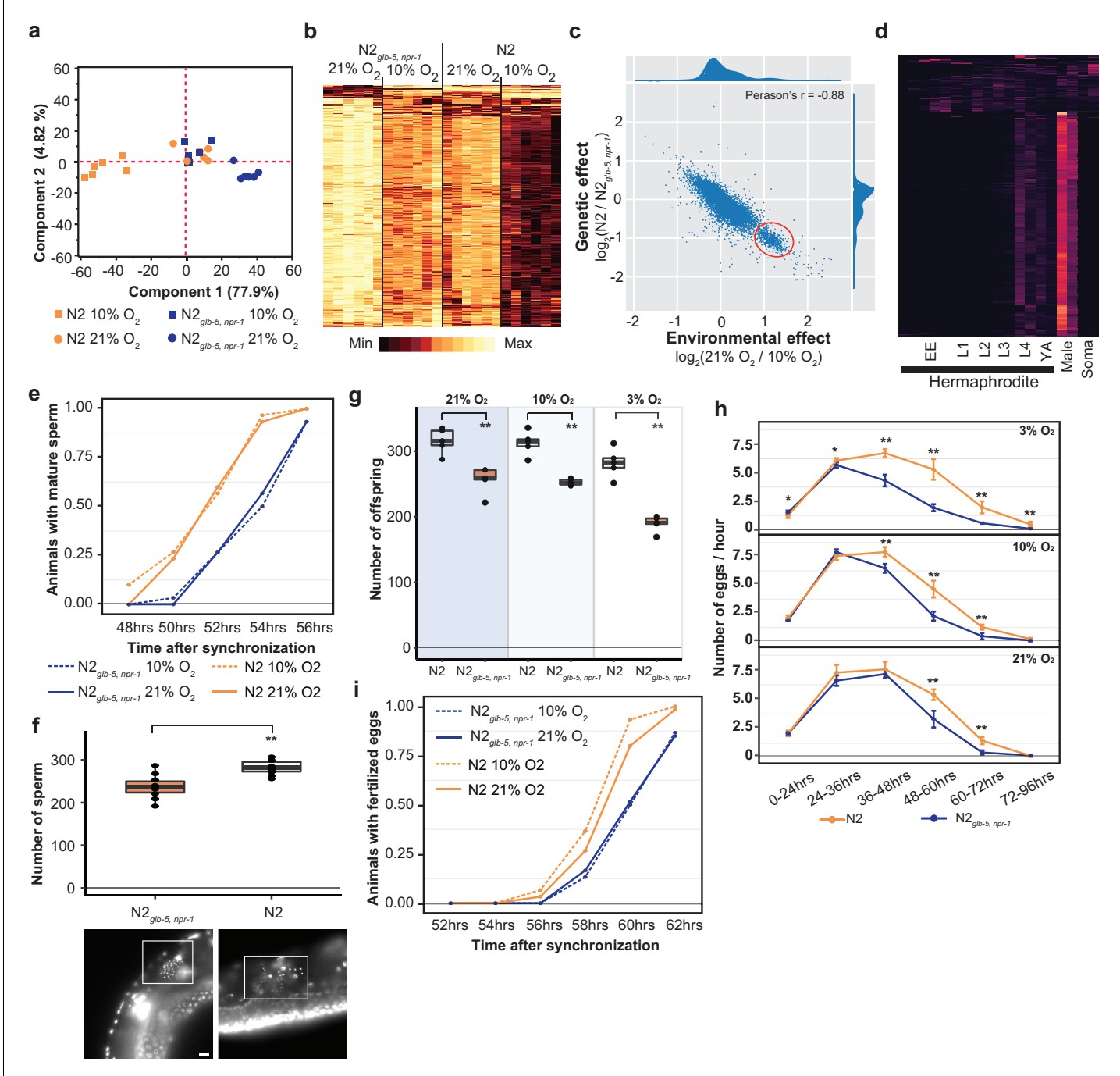

**Figure 5.** Reproductive timing in N2 occurs earlier than the N2$_{glb-5,npr-1}$ strain. (**a**) PCA analysis of transcriptional profiles of bleach-synchronized N2 and N2$_{glb-5,npr-1}$ animals grown in 10% or 21% environmental $O_2$ (six replicates per strain/condition). The largest two eigenvectors are shown, along with the amount of variance they explain. Developmental age of animals is approximately L4 stage. (**b**) Hierarchical clustering of normalized, differentially expressed genes. Columns show strain and conditions; rows show gene expression. (**c**) Averaged effect of genotype (y-axis) vs environment (x-axis) for each gene (**Supplementary file 1**). A small cluster of 652 genes with similar changes is circled in red. (**d**) The developmental expression of these 652 genes was further investigated using a previously published dataset. Columns show developmental stage and rows show each gene. Most of these gene peaked in expression in L4 hermaphrodite animals and was further enriched in male L4 animals (Male). Soma indicates expression levels from somatic cells, suggesting this cluster is enriched in germline cells. (**e**) Animals identified with mature sperm. x-axis indicates time since synchronization using hatch-off. Strain/condition shown in legend. p=0.0076 by Friedman test. (**f**) Number of sperm produced by each strain as determined by DAPI straining. Representative images are shown below. Scale bar = 10 μm. **p<*0.01* by Wilcoxon-Mann-Whitney nonparametric test. (**g**) Averaged total number of offspring produced by each strain when grown in different environmental $O_2$ levels. **p<*0.01* by Wilcoxon-Mann-Whitney nonparametric

*Figure 5 continued on next page*

*Figure 5 continued*

test. (h) Averaged egg-laying rate of L4-synchronized N2 and N2$_{glb-5,npr-1}$ animals when grown at different O$_2$ levels. x-axis indicates time since L4 stage. NS, not significant, *p<0.05, **p<0.01 by Wilcoxon-Mann-Whitney nonparametric test. (i) Number of animals observed with fertilized eggs in their uterus. x-axis indicates time from synchronized egg-lay. p=0.0109 by Friedman test.

DOI: https://doi.org/10.7554/eLife.38675.023

The following source data is available for figure 5:

**Source data 1.** List of normalized differentially expressed genes for PCA analysis and Hierarchical clustering.

DOI: https://doi.org/10.7554/eLife.38675.024

**Source data 2.** List of the relative expression levels of protein coding genes across all of the developmental stages highlighted in *Figure 5c*.

DOI: https://doi.org/10.7554/eLife.38675.025

**Source data 3.** Number of animals identified with mature sperm at indicated timepoint shown in *Figure 5e*.

DOI: https://doi.org/10.7554/eLife.38675.026

**Source data 4.** Number of sperm produced by each strain as determined by DAPI straining shown in *Figure 5f*.

DOI: https://doi.org/10.7554/eLife.38675.027

**Source data 5.** Mean number of offspring produced by each strain when grown in different environmental O$_2$ levels shown in *Figure 5g*.

DOI: https://doi.org/10.7554/eLife.38675.028

**Source data 6.** Mean egg-laying rate of L4-synchronized N2 and N2$_{glb-5,npr-1}$ animals when grown at different O$_2$ levels shown in *Figure 5h*.

DOI: https://doi.org/10.7554/eLife.38675.029

**Source data 7.** Number of animals observed with fertilized eggs in their uterus at indicated timepoint shown in *Figure 5i*.

DOI: https://doi.org/10.7554/eLife.38675.030

results further support that social behavior is not the major determinant of fitness levels in laboratory conditions.

## Development speed and spermatogenesis are increased in N2 in an O$_2$-independent manner

To gain more insight into the phenotypes that could be responsible for the fitness increases of the N2 strain, we performed RNA sequencing to analyze the transcriptomes of bleach-synchronized N2 and N2$_{glb-5, npr-1}$ animals grown in either 10% O$_2$ or 21% ambient O$_2$ levels. Animals were allowed to develop to the L4 stage and harvested at identical times. We first performed Principal Component Analysis (PCA) analysis on differentially expressed genes to analyze how the environmental and genetic differences globally regulated the transcriptomes of the animals. If environmental O$_2$ and the genetic background had independent effects on the transcriptomes, we expected to find two major components in the PCA analysis. However, the PCA analysis identified a single component that explained the majority of the variance (77.9%). The genetic and environmental perturbations had similar effects on the first component in an additive manner (*Figure 5a*). Reducing O$_2$ levels from 21% to 10% had similar effects on the transcription profiles as changing the background from N2$_{glb-5, npr-1}$ to N2. Consequently, the animals that differed in both genetic background and environmental O$_2$ levels (N2–21% O$_2$ vs N2$_{glb-5, npr-1}$–10% O$_2$) also showed the most similar transcriptional profiles. These patterns were also seen in Hierarchical Clustering using the 1202 differentially expressed genes (*Figure 5b*). These results suggest that the foraging behavioral differences are not responsible for the underlying transcriptomics differences between the different strains and environmental conditions.

The effects of the derived *npr-1* and *glb-5* alleles mimics the effects of lowering environmental O$_2$ from 21% to 10%. To further gain insight into this connection, we plotted the average transcriptional change between the strain backgrounds vs the average transcriptional change between the environmental O$_2$ concentrations for each gene (*Figure 5c*, *Supplementary file 1*). Surprisingly, we observed a bimodal distribution of values, with a cluster of 652 genes centered at 1.2 log$_2$-fold change (*Figure 5c* – red circle). This is unexpected, as it suggests that the environmental and genetic perturbations had identical effects on transcription for all these genes. When we inspected this list of genes, we noticed a large number of genes that are known to be involved in spermatogenesis. We further investigated the developmental regulation of these 652 genes using previously published transcriptomics data isolated from hermaphrodites or males at specific developmental time points (*Boeck et al., 2016*) (*Figure 5d*). The expression of the majority of these genes peaked during the L4 stage in hermaphrodites, was further enriched in L4 males, and suppressed in somatic cells

isolated from L4 animals. These observations are consistent with this cluster of genes being involved in spermatogenesis, which occurs during the L4 stage (when RNA was isolated) in hermaphrodite animals.

We reasoned that the transcriptomics data could indicate a difference in the relative timing of spermatogenesis and/or the number of sperm that are produced in each genetic background/environmental condition. L1 larval stage animals were synchronized; subsequent differences in developmental speed would result in animals in slightly different stages of L4. To test this, we synchronized N2$_{glb-5,\ npr-1}$ and N2 animals, placed them in 10% or 21% environmental $O_2$, and identified the number animals containing mature sperm at 2 hr intervals from 48 to 56 hr. N2 animals began spermatogenesis approximately 2 hr earlier than the N2$_{glb-5,\ npr-1}$ animals, regardless of the environmental $O_2$ levels (*Figure 5e*). Hermaphrodites undergo spermatogenesis for a fixed period of time before permanently switching gametogenesis to the production of oocytes, resulting in the development of a fixed number of self-sperm that are stored in the spermathecae (*Hubbard and Greenstein, 2005*). To test whether these strains produced the same number of sperm, we used DAPI staining to count the number of sperm found in the spermathecae. Not only did N2 animals start spermatogenesis earlier, they also produced more sperm (*Figure 5f*). The total fecundity of N2 hermaphrodites that do not mate with males is determined by the number of self-sperm. We confirmed that the difference in self-sperm number also resulted in a larger overall brood size (*Figure 5g*) and as expected from computational modeling (*Large et al., 2017*), an increased rate of egg-laying later on in life (*Figure 5h*).

The timing of sexual maturity is an important factor in determining the fitness of animals. We also tested whether the differences in timing of spermatogenesis could lead to differences in when fertilized eggs are produced. We performed similar experiments as above and monitored the time fertilized eggs could be observed in the uterus at two-hour intervals. Again, we observed a difference in N2 and N2$_{glb-5,\ npr-1}$ animals at both 10% and 21% environmental $O_2$ levels. N2 animals were observed to contain fertilized eggs approximately 1 hr earlier that N2$_{glb-5,\ npr-1}$ animals (*Figure 5i*). The difference in timing of spermatogenesis and fertilization (2 hr vs 1 hr), potentially reflects the fact that N2 animals produce more sperm before switching to oogenesis.

These experiments suggest that the differences in transcription between N2 and N2$_{glb-5,\ npr-1}$ could be caused by differences in sexual maturity. We are unable, however, to explain the differences in transcription we observed between 10% and 21% $O_2$ as mature sperm was observed at similar times in these different environmental conditions (*Figure 5e*). Potentially, the rate of spermatogenesis or expression levels of genes are modified by $O_2$ levels that are not reflected in the timing of the presence of mature sperm.

## Derived alleles of *npr-1* and/or *glb-5* increase food consumption in an $O_2$-independent manner

Life-history tradeoffs have been proposed in evolutionary theory to account for the linkage between two different traits. Assuming an individual can acquire a finite amount of energy, the investment of energy into one trait leads to consequential changes in other traits as energy resources are shunted into different directions. For example, artificial selection experiments on early fecundity in *C. elegans* resulted in decreased reproduction late in life (*Anderson et al., 2011*). The N2 strain seems to violate this tradeoff, as it sexually matures earlier than N2$_{glb-5,\ npr-1}$, but also produces more eggs later on in life. We measured the size of N2 and N2$_{glb-5,\ npr-1}$ animals and found that N2 animals were also larger than N2$_{glb-5,\ npr-1}$ animals at synchronized time points (*Figure 6a*). These observations suggest that the assumption of a fixed energy acquisition for N2$_{glb-5,\ npr-1}$ and N2 might be violated. This would be consistent with Andersen et al's observation that metabolism genes were upregulated by the derived *npr-1* allele, which they proposed represented differences in food intake (*Andersen et al., 2014*). It would also be consistent with the role of orthologs of *npr-1* in other species. *npr-1* encodes an ortholog to neuropeptide Y receptors, which are reported to regulate feeding behavior in fishes, birds, and mammals (*Ando et al., 2001*; *Lecklin et al., 2002*; *Matsuda, 2009*).

To test this hypothesis, we first utilized a previously described feeding assay to measure the ability of a strain to clear *E. coli* OP50 bacteria from liquid S-media (*Gomez-Amaro et al., 2015*). In this assay, individual wells are seeded with a defined number of bacteria and 20 worms. Each day, the optical density of each well is measured to estimate the amount of food consumed by the worms. In

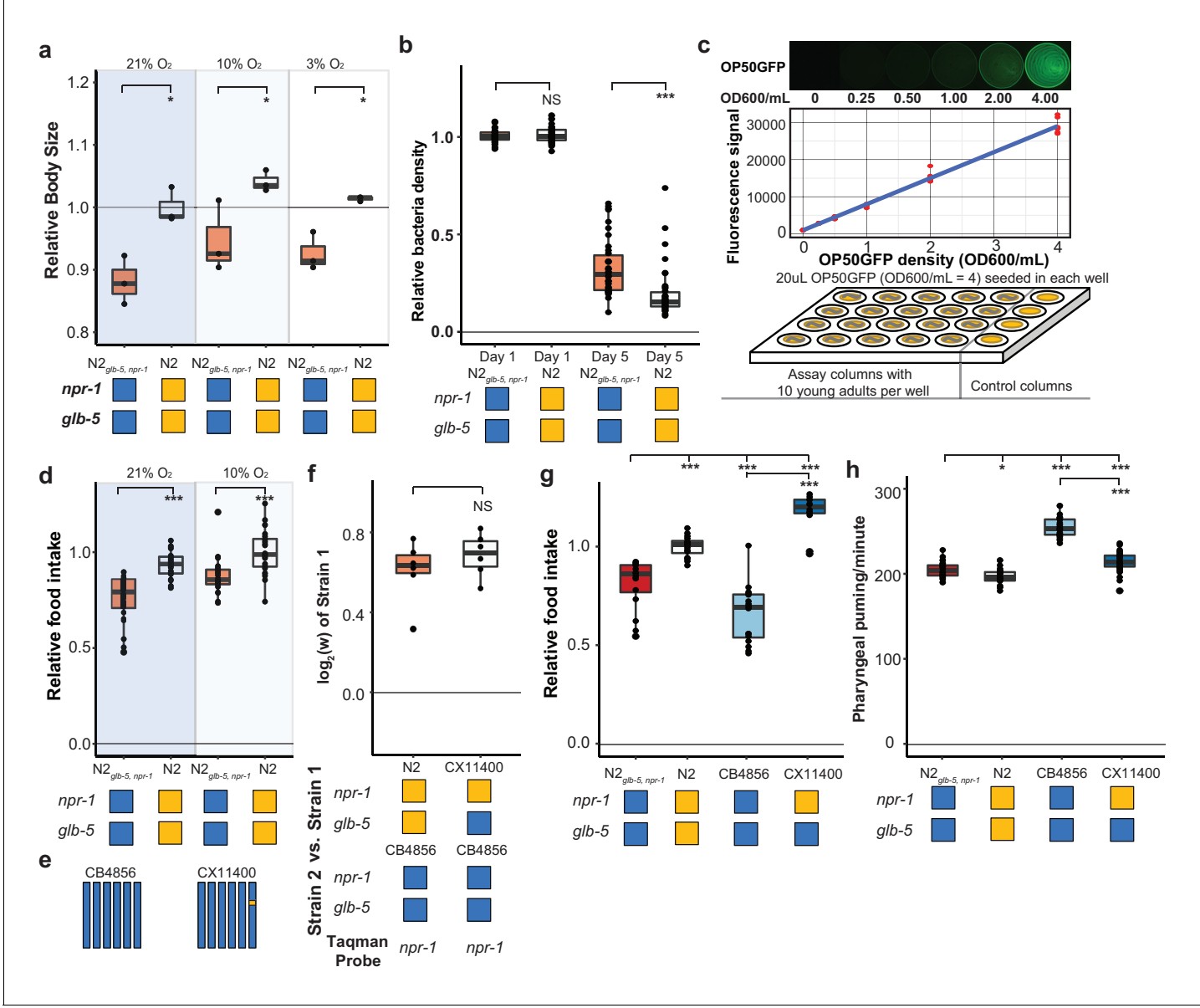

**Figure 6.** Feeding differences of strains containing derived alleles. (a) N2 and N2_{glb-5,npr-1} animals were synchronized by hatch-off and allowed to grow at the indicated O₂ levels for 72 hr. Video recordings were used to estimate the size of the animals. *p<0.05 by Wilcoxon-Mann-Whitney nonparametric test. (b) A previously published liquid, bacterial clearing assay was used to estimate food consumption for the N2_{glb-5,npr-1} and N2 animals. On day 4, N2 animals had consumed more bacteria than N2_{glb-5,npr-1}animals. NS, not significant, ***p<0.001 by Wilcoxon-Mann-Whitney nonparametric test. (c). To test food consumption on agar plates, we developed a new assay by seeding 24-well agar plates with defined amounts of OP50-GFP bacteria. The number of bacteria on the plate could be estimated using a microplate reader. (d) N2 animals consumed more food than N2_{glb-5,npr-1} regardless of foraging behaviors. ***p<0.001 by Wilcoxon-Mann-Whitney nonparametric test. (e) Schematic of CB4856 wild strain (blue) and a NIL (CX11400) containing the N2 allele of *npr-1* from N2 (orange). (f) We tested the fitness effect of the N2 allele of *npr-1* in the CB4856 wild strain using the CX11400 NIL strain. NS, not significant by Wilcoxon-Mann-Whitney nonparametric test. (g) Food consumption assays between CB4856 and N2 strains or CB4856 and the CX11400 NIL. ***p<0.001 by ANOVA with Tukey's Honest Significant Difference test. (h) Pharyngeal pumping rates of N2, CB4856 and two NIL strains. *p<0.05, ***p<0.001 by ANOVA with Tukey's Honest Significant Difference test.

DOI: https://doi.org/10.7554/eLife.38675.031

The following source data and figure supplements are available for figure 6:

**Source data 1.** Growth rates of N2 and N2_{glb-5,npr-1} shown in *Figure 6a*.
DOI: https://doi.org/10.7554/eLife.38675.034
**Source data 2.** Food consumption in liquid S media shown in *Figure 6b*.
DOI: https://doi.org/10.7554/eLife.38675.035

*Figure 6 continued on next page*

*Figure 6 continued*

**Source data 3.** Food consumption assay setup using OP50 GFP on 24-well agar plate shown in *Figure 6c*.
DOI: https://doi.org/10.7554/eLife.38675.036
**Source data 4.** N2 animals consume more food in $O_2$ independent manner shown in *Figure 6d*.
DOI: https://doi.org/10.7554/eLife.38675.037
**Source data 5.** Fitness effect of the N2 allele of *npr-1* in the CB4856 wild strain shown in *Figure 6f*.
DOI: https://doi.org/10.7554/eLife.38675.038
**Source data 6.** Food consumption assays between CB4856 and N2 strains or CB4856 and the CX11400 NIL shown in *Figure 6g*.
DOI: https://doi.org/10.7554/eLife.38675.039
**Source data 7.** Pharyngeal pumping rates of N2, CB4856 and two NIL strains shown in *Figure 6h*.
DOI: https://doi.org/10.7554/eLife.38675.040
**Figure supplement 1.** Measurement of pharyngeal sizes of adult animals.
DOI: https://doi.org/10.7554/eLife.38675.032
**Figure supplement 1—source data 1.** Measurement of pharyngeal sizes of adult animals shown in *Figure 6—figure supplement 1b*.
DOI: https://doi.org/10.7554/eLife.38675.033

these conditions, N2 cleared the bacteria faster than N2$_{glb-5, npr-1}$ animals (*Figure 6b*). While these assays supported our hypothesis, liquid media is fundamentally different from the conditions experienced on agar plates, making it difficult to generalize the results from one condition to the other. To this end, we developed a new food consumption assay on agar media in 24-well plates. In this assay, each well was seeded with a defined amount of OP50-GFP, which we found could be quantified in a linear manner using a plate reader (*Figure 6c*). When we tested N2 and N2$_{glb-5, npr-1}$ animals in 10% or 21% environmental $O_2$ levels, we found N2 consumed more food than N2$_{glb-5, npr-1}$ in both environmental conditions (*Figure 6d*). Interestingly, we found animals grown in 10% $O_2$ also consume more food than animals grown in 21% $O_2$. These experiments indicate that N2 animals consume more food than N2$_{glb-5, npr-1}$.

We next decided to test whether the derived allele of *npr-1* could increase the fitness and feeding rate in a different genetic background. We used the CB4856 wild strain isolated from pineapple fields in Hawaii, which has relatively low relative fitness in laboratory conditions (*Figure 4b*), taking advantage of a previously constructed NIL of *npr-1* introgressed from N2 into the CB4856 background (CX11400) (*Bendesky et al., 2012*) (*Figure 6e*). We found that the N2 region surrounding *npr-1* also conferred a fitness advantage in the CB4856 background (*Figure 6f*). The estimated selective coefficients of the derived allele of *npr-1* was higher in the CB4856 background than the N2 background (s = 0.61 vs s = 0.30), potentially due to the lower relative fitness of the CB4856 strain. The food consumption of these strains was consistent with the fitness differences (*Figure 6g*). The derived allele of *npr-1* increased food consumption in both genetic backgrounds but its effect was higher in CB4856.

Food is consumed from the environment by the periodic contraction and relaxation of the pharyngeal muscle which serves to bring material from the environment into the pharynx and filter out bacterial cells (*Fang-Yen et al., 2009*). To test whether the increase in food consumption could be explained by an increase in the rate of pumping, we measured the pharyngeal pumping rate of the N2$_{glb-5, npr-1}$, N2, CB4856, and CX11400 strains. The effects of the derived allele of *npr-1* was epistatic with respect to the N2 or CB4856 background. The derived allele decreased the pumping rate in the CB4856 background but had no effect in the N2 background (*Figure 6h*). The effect of the derived allele of *npr-1* on pumping rate is surprising. Pumping rate is often used as a proxy for food consumption; our results indicate that increased pharyngeal pumping does not necessarily lead to increases in food consumption.

We also measured a number of size parameters of the pharynx but found no obvious differences that could account for the increased food consumption (*Figure 6—figure supplement 1*). Potentially, the pharynx is more efficient at bringing food in from the external environment due to stronger pump strength, more efficient filtering processes or other unknown behavioral differences that contribute to food intake.

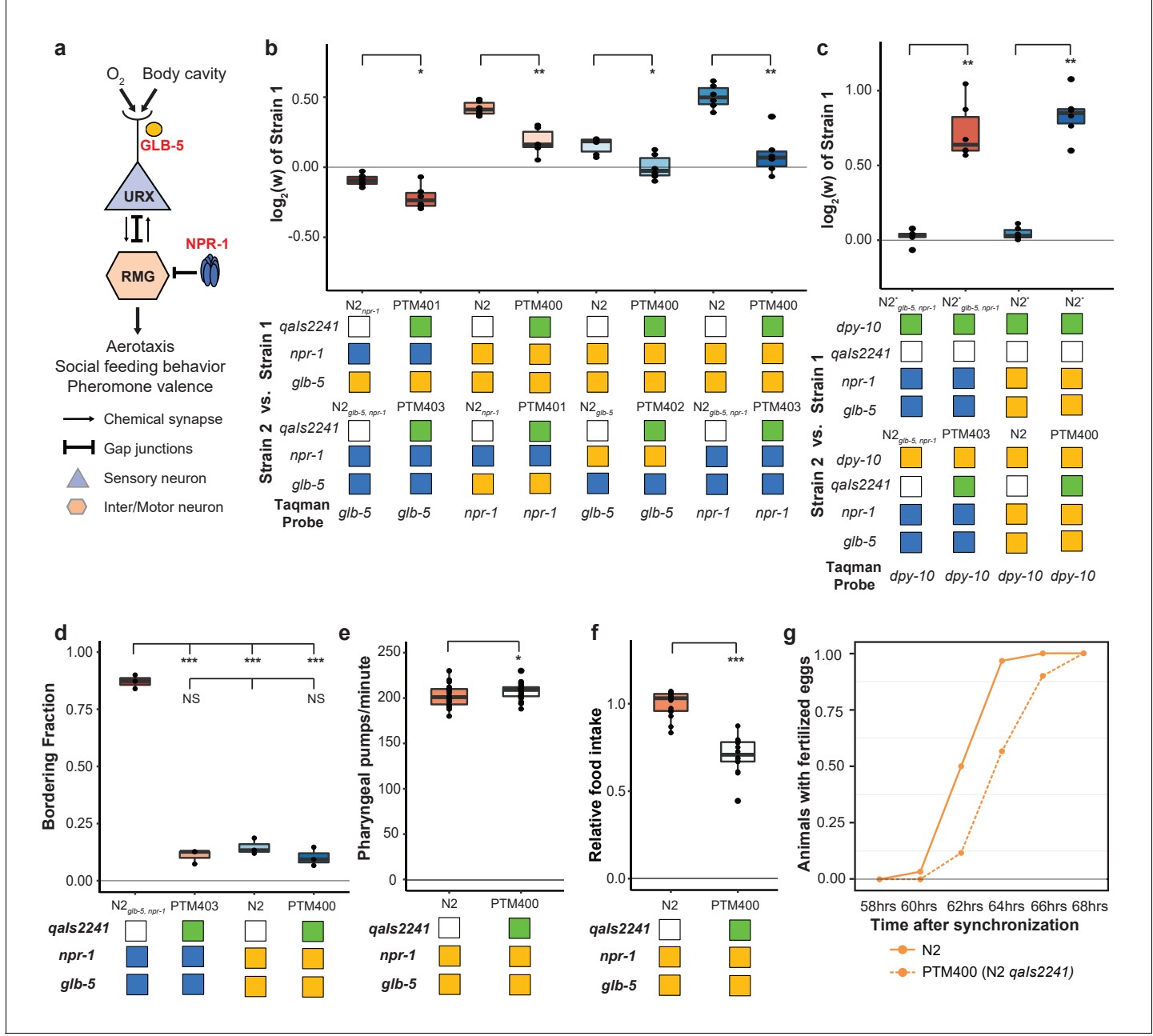

**Figure 7.** $O_2$-sensing neurons contribute to fitness differences of N2 and $N2_{glb-5,npr-1}$. (**a**) Schematic showing putative cellular sites of action for *glb-5* and *npr-1*. *glb-5* modulates $O_2$ responses in the URX body cavity neurons. *npr-1* is thought to modulate electrical signaling in the RMG hub-and-spoke neuron which forms gap junctions onto URX. (**b** and **c**) Competition experiments between indicated strains. *qals2241* is an integrated genetic cassette that ablates the URX, AQR, and PQR neurons. Green indicates the presence of the cassette (and loss of URX, AQR, and PQR neurons). *p<0.05, **p<0.01 by Wilcoxon-Mann-Whitney nonparametric test. (**d**) Bordering rates of indicated strains. The *qals2241* cassette suppresses bordering of the $N2_{glb-5,npr-1}$ strains. NS not significant, *p<0.05 by Wilcoxon-Mann-Whitney nonparametric test. (**e**) Pharyngeal pumping rates of N2, and N2 strains carrying the *qals2241* cassette. *p<0.05 by Wilcoxon-Mann-Whitney nonparametric test. (**f**) Relative food consumption rates between the indicated strains. ***p<0.001 by Wilcoxon-Mann-Whitney nonparametric test. (**g**) Number of animals observed with fertilized eggs in their uterus. x-axis indicates time from synchronized egg-lay. p=0.0455 by Friedman test.

DOI: https://doi.org/10.7554/eLife.38675.041

The following source data is available for figure 7:

**Source data 1.** Competition experiments between indicated strains carry *qals2241* cassette shown in *Figure 7b*.
DOI: https://doi.org/10.7554/eLife.38675.042
**Source data 2.** Competition experiments between indicated strains carry *qals2241* cassette shown in *Figure 7c*.

*Figure 7 continued on next page*

*Figure 7 continued*

DOI: https://doi.org/10.7554/eLife.38675.043

**Source data 3.** Bordering fraction of indicated strains shown in *Figure 7d*.

DOI: https://doi.org/10.7554/eLife.38675.044

**Source data 4.** Pharyngeal pumping rates of N2 and N2 carries *qaIs2241* cassette shown in *Figure 7e*.

DOI: https://doi.org/10.7554/eLife.38675.045

**Source data 5.** Food consumption assay of N2 and N2 carries *qaIs2241* cassette shown in *Figure 7f*.

DOI: https://doi.org/10.7554/eLife.38675.046

**Source data 6.** Number of animals observed with fertilized eggs in their uterus shown in *Figure 7g*.

DOI: https://doi.org/10.7554/eLife.38675.047

## Fitness gains of the derived alleles require the URX, AQR, and/or PQR neurons

We next decided to gain insight into the cellular mechanisms by which *npr-1* and *glb-5* increased fitness of the strains. Previous studies have shown that *npr-1* and *glb-5* regulate social behavior through the URX-RMG neuronal circuit (*Figure 7a*). *glb-5* tunes $O_2$-sensitivities of the URX oxygen-sensing neuron pair through regulation of $O_2$-sensing guanylyl cyclases, leading to changes in influx of $Ca^{++}$ into the cell body (*Abergel et al., 2016*; *Gross et al., 2014*; *McGrath et al., 2009*; *Oda et al., 2017*; *Persson et al., 2009*). The derived allele of *npr-1* inhibits the activity of the RMG hub interneuron which suppresses aerotaxis and social behavior (*Laurent et al., 2015*; *Macosko et al., 2009*). The RMG neurons connect to URX and a number of other sensory neurons through gap junctions, which are necessary for foraging behaviors (*Jang et al., 2017*). URX neurons also integrate $O_2$ with internal nutrient reserves (*Witham et al., 2016*). To test the role of URX in the fitness gains of the *npr-1* and *glb-5* derived alleles, we used the *qaIs2241* integrated cassette that specifically kills the $O_2$-sensing neurons URX, AQR and PQR (*Chang et al., 2006*). We crossed this cassette into the N2$_{npr-1}$, N2$_{glb-5}$ and N2$_{glb-5, npr-1}$ strains and repeated the pairwise competition experiments performed in *Figure 2a* using strains that now also contained the *qaIs2241* cassette. In all cases, the relative fitness gains of the derived alleles were decreased by the presence of the neuronal ablation (*Figure 7b*).

These experiments suggest that the derived alleles either activate or disinhibit the URX, AQR, and or PQR neurons which leads to increases in fitness. To distinguish between these possibilities, we competed N2 and N2$_{glb-5, npr-1}$ strains with and without the *qaIs2241* against each other. Strains that carried the *qaIs2241* cassette were dramatically less fit than the control worms, suggesting that URX, AQR, and PQR promote fitness in laboratory conditions (*Figure 7c*).

We and others have shown that *glb-5* and *npr-1* are pleiotropic, regulating social behavior and food consumption. Potentially this pleiotropy arises from the ability of the URX, AQR, and PQR neurons to these biological traits. To test this, we phenotyped strains that carried the *qaIs2241* cassette for social behaviors, food consumption and reproductive timing (*Figure 7d–g*). These experiments indicated that these neurons are required for each of these three traits. Interestingly, food consumption in the *qaIs2241* strains was reduced without a corresponding change in pharyngeal pumping rate, further confirming that these phenotypes could be separated from each other at a genetic and cellular level.

## Fitness gains, increased food consumption, and earlier reproductive timing in N2 require the *daf-22* gene

We also decided to test whether ascaroside pheromones were necessary for the fitness differences between N2 and N2$_{glb-5,npr-1}$. Nematodes release a number of ascaroside molecules, which are in turn sensed by a distributed neural circuit that integrates and modifies a number of behavioral and developmental phenotypes (*Butcher, 2017*; *Ludewig and Schroeder, 2013*). There are a few reasons to think that ascaroside pheromones might be involved in the fitness gains of the N2 strain. First, work by Andersen et. al indicated that population density directly impacts lifetime fecundity and adult body length differences between N2 and CB4856 strains (*Andersen et al., 2014*). Second, our previous studies of *C. elegans* domestication to liquid cultures has found that pheromone signaling was modified by fixed genetic changes (*Large et al., 2016*; *McGrath et al., 2011*). Finally, the

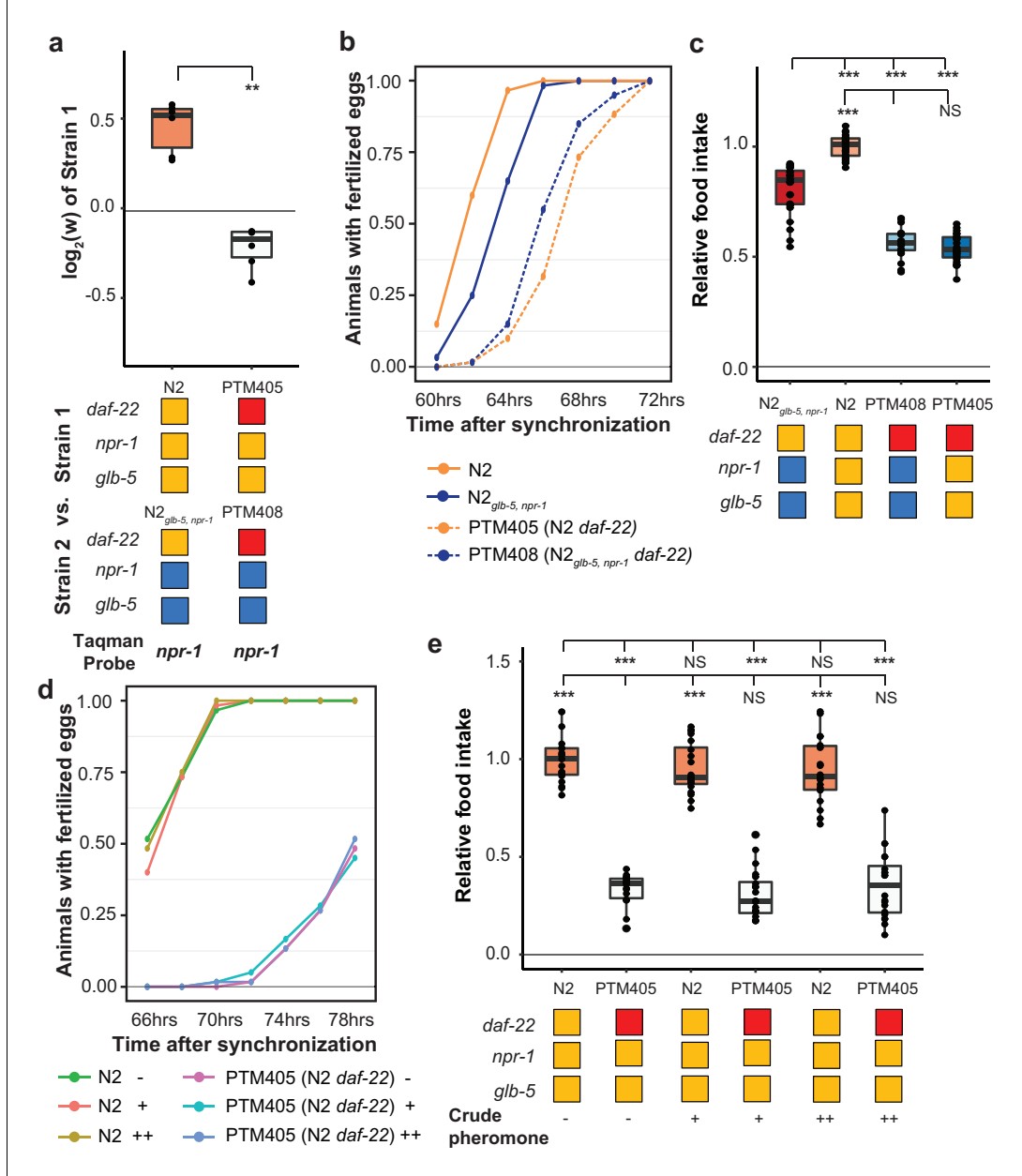

**Figure 8.** *daf-22* is required for fitness differences of N2 and N2$_{glb-5,npr-1}$. (a) Competition experiments between indicated strains. *daf-22* encodes a sterol carrier protein, which is required for biosynthesis of most ascaroside pheromones. Red indicates the strain contain a deletion that spans the gene. **p<0.01 by Wilcoxon-Mann-Whitney nonparametric test. (b) Number of animals that carry fertilized eggs at the indicated timepoints. p=6.61×10$^{-4}$ by Friedman test. (c) On plate feeding assays of the indicated strains. NS, not significant, ***p<0.001 by ANOVA with Tukey's Honest Significant Difference test. (d and e) Attempts to rescue the relative food intake and reproductive timing defects of the *daf-22* strain using crude pheromone. Neither of two concentrations of crude pheromone isolated from animals grown in liquid cultures had a significant effect on the two traits. NS, not significant, ***p<0.001 by ANOVA with Tukey's Honest Significant Difference test. p=7.45×10$^{-6}$ by Friedman test.

DOI: https://doi.org/10.7554/eLife.38675.048

The following source data is available for figure 8:

**Source data 1.** Competition experiment between indicated strains shown in *Figure 8a*.

DOI: https://doi.org/10.7554/eLife.38675.049

**Source data 2.** Number of animals observed with fertilized eggs in their uterus shown in *Figure 8b*.

DOI: https://doi.org/10.7554/eLife.38675.050

**Source data 3.** Food consumption assay of indicated strains shown in *Figure 8c*.

DOI: https://doi.org/10.7554/eLife.38675.051

*Figure 8 continued*

**Source data 4.** Number of animals observed with fertilized eggs in their uterus in different pheromone concentration shown in *Figure 8d*.
DOI: https://doi.org/10.7554/eLife.38675.052
**Source data 5.** Food consumption assay in different pheromone concentration shown in *Figure 8e*.
DOI: https://doi.org/10.7554/eLife.38675.053

derived alleles of *npr-1* and *glb-5* have been shown to modify pheromone valence in a variety of contexts (*Fenk and de Bono, 2017*; *Jang et al., 2012*; *Macosko et al., 2009*; *Oda et al., 2017*). To test the role of ascaroside pheromones, we followed previous publications using a genetic knockout of the *daf-22* gene, which encodes a peroxisomal enzyme required for the biosynthesis of *C. elegans* pheromones (*Butcher et al., 2009*) and accumulation of lipid droplets (*Zhang et al., 2010*), using CRISPR-Cas9 enabled genome editing to create a large deletion of *daf-22* in the N2 strain, which was then crossed to the N2$_{glb-5,npr-1}$ background. Competition experiments demonstrated that *daf-22* was necessary for the fitness advantage of derived *npr-1* and *glb-5* alleles (*Figure 8a*). In addition, *daf-22* was necessary for the faster sexual maturity (*Figure 8b*) and increased food intake (*Figure 8c*) of the N2 strain compared to N2$_{glb-5,npr-1}$. These data suggest that *npr-1* and *glb-5* reprogram pheromone responses resulting in increased sexual maturity and ability to consume food.

*daf-22* encodes a peroxisomal fatty acid β-oxidation enzyme. Besides its role in the biosynthesis of ascaroside pheromones, *daf-22* has recently been shown to play a distinct role in ASK neurons, where it is required for the metabolization of fatty acids that stimulate the endoplasmic reticulum stress response, promoting the transcription of insulin-like peptides that regulate dauer formation and other biological processes (*Park and Paik, 2017*). *daf-22* mutants also accumulate massive amounts of fatty acids and fatty acyl-CoAs in their intestines (*Butcher et al., 2009*; *Joo et al., 2009*; *Li et al., 2016*), which can potentially regulate feeding behavior through homeostasis mechanisms (*Hyun et al., 2016*). To determine if the differences observed in the *daf-22* mutants were caused by the lack of ascaroside pheromones, we attempted to rescue these phenotypes using two concentrations of crudely purified pheromones isolated from animals grown in liquid cultures. Neither of these concentrations were able to rescue the differences in food consumption or reproductive timing (*Figure 8d,e*). These experiments suggest that the effects of the *daf-22* mutants we have observed might be independent of their role in producing ascaroside pheromones.

## Discussion

In this report, we studied the fitness consequences of two derived alleles that arose and fixed in the N2 strain after isolation from the wild. We find that both alleles can be adaptive, with selective coefficients that are larger than many characterized beneficial alleles from other species. These results are consistent with the derived alleles spreading through the ancestral N2 populations due to positive selection. If this was true, it would suggest that the derived allele of *npr-1* arose first, as the derived *glb-5* allele is only beneficial in this derived genetic background. However, the demographic history and laboratory environment of how N2 was grown at the time these alleles arose is largely lost (*Sterken et al., 2015*). The exact laboratory growth conditions (liquid axenic vs. solid media), transfer processes (picking vs. chunking) and effective population sizes (between 4 and 1000) used to propagate a *C. elegans* strain is incredibly variable. It is likely that the evolutionary forces responsible for the fixation of these alleles will remain lost to history.

Nevertheless, the ability of positive selection to act upon the derived *npr-1* allele can be observed in current experiments. A recent example is provided by Noble and colleagues, who created a large mapping population between 16 parental strains (including N2 and CB4856) to create a large panel recombinant inbred lines (RILs) (*Noble et al., 2017*). During the outcrossing phase of construction, the N2 allele of *npr-1* spread through the population to fixation, consistent with its dominant action and the strong selective advantage of this allele. Potentially, variation in *npr-1* affected allele frequencies of unlinked loci as well. For example, an excess of CB4856 haplotypes was observed in the RILs, suggesting that CB4856 haplotypes were more likely to contain beneficial alleles. Our measurements of the relative fitness of the CB4856 strain, however, creates an apparent paradox, as CB4856 was one of the least fit strains among the wild strains we tested (*Figure 4b*). Potentially, epistatic interactions between CB4856 alleles and the derived allele of *npr-1* could help resolve this; the

effect of *npr-1* on food intake and fitness is higher in the CB4856 background (*Figure 6f,g*). Differences in effect size of a focal allele in different genetic backgrounds is considered evidence for the existence of epistasis (*Gibson and Dworkin, 2004*). Potentially, the presence of laboratory-derived alleles in mapping populations will skew not only the allele frequencies of these beneficial alleles, but also natural genetic variants that interact epistatically with them.

Evolution of behavioral traits is one strategy for animals to respond to a new environment. The identification of a polymorphism in *npr-1* has served as an example of how behavioral variation can arise from genetic variation. However, our work suggests that the social/solitary feeding behavioral changes of N2 are not sufficient for explaining its fitness gains. Rather, we propose that changes to food intake, sexual maturity, and fecundity are more important. One unresolved question is why wild strains do not eat as much food as the N2 strain. We believe there must be some sort of tradeoff – either energetically or developmentally – that makes the derived mutation unfavorable in their natural environments. Mechanistic understanding of the energetic forces necessary for *C. elegans* to bring food into their pharynx is lacking. In fact, pharyngeal pumping rates are often used as proxies for food intake, which we have shown here can be unrelated to the amount of food consumed. Potentially, the thick slurry of food in laboratory plates is completely different biophysically from the mixed bacterial species encountered on rotting material in the wild. Alternatively, differences in feeding behaviors unrelated to social/solitary behaviors might also mediate the differences in food intake. Our experiments suggest that previously described roaming/dwelling differences are also not responsible, however, additional uncharacterized behavioral differences could influence food intake.

The changes to fitness and food consumption in the N2 strain appear to be mediated by the nervous system, which we propose occurs through changes in the function and/or downstream effects of the URX sensory neurons. In this paper, we have also shown that animals that lack the URX neurons consume food at lower rates. How does URX modify food consumption? One possibility is URX regulates pharyngeal neurons extrasynaptically through neuropeptides or through chemical synapses onto the RIP interneurons, which represent the only connection between the somatic and pharyngeal nervous system. Alternatively, URX could regulate food consumption indirectly by stimulating metabolism of fatty acids. URX, along with AQR and PQR, are body cavity neurons, sending ciliated dendrites into the coelomic fluid, which serves as the circulatory system for *C. elegans* (*White et al., 1986*). Besides sensing external $O_2$, URX neurons monitor fat stores, which are thought to regulate tonic $Ca^{++}$ responses of the URX neurons (*Witham et al., 2016*). URX, in turn, can stimulate fat loss, creating a homeostatic loop that ensures that fat mobilization only occurs when there are sufficient fat reserves and when environmental $O_2$ is high enough to metabolize the fatty acids into energy (*Witham et al., 2016*). The changes to the N2 strain could have resulted in URX triggering fatty acid metabolism at a higher rate in laboratory conditions. The access energy could be used to speed development and increase growth. Why, then, would the animals consume more food? Fat metabolism has been shown to regulate satiety behavior in *C. elegans*, which could account for increases in food consumption we see in these strains (*Hyun et al., 2016*).

This model could also explain the effects we see in the *daf-22* mutants, which we originally explored for the potential role of pheromone responses in feeding and fitness changes. *daf-22* mutants accumulate large amounts of fatty acids in their bodies, which potentially inhibits the food consumption rates of these animals. However, our experiments do not preclude a role for pheromones in these fitness and food consumption changes to N2. Potentially, our crude purifications do not capture physiologically relevant levels and ratios of the complex pheromone mixtures. Pheromones might also contribute to these differences in combination with other *daf-22*-dependent pathways. Primer pheromones have been shown to influence body fat metabolism in *C. elegans* through the ADL sensory neuron (*Hussey et al., 2017*). ADL sensory neurons are regulated by pheromones in an *npr-1*-dependent manner (*Abergel et al., 2016*; *Fenk and de Bono, 2017*; *Jang et al., 2012*; *Jang et al., 2017*; *Macosko et al., 2009*). It is possible these changes, or other parts of the pheromone circuits are also necessary. Future experiments, enabled by the development of the on-plate food consumption assay, should be enlightening.

Our work underscores issues with growing organisms in the laboratory for multiple generations. Despite the attempts of researchers to create fertile conditions for nematodes to grow in, we found a large difference in relative fitness between different strains of *C. elegans* when competed in the laboratory. Natural genetic variation and de novo variation both result in fitness differences that selection can act on. Experimenters using wild strains of nematodes must take care in designing

experiments to account for this, especially in wild strains with lower initial fitness levels. We believe that the laboratory selection pressures we characterized here will generalize to other invertebrate and vertebrate animals. If so, the behaviors and physiology of these animals will also be modified over generations of growth. Our work suggests that not only will the traits that confer fitness advantages be modified, but potentially additional traits due to the pleiotropic actions of many genes, and relaxed stabilizing selection on traits in laboratory conditions.

# Materials and methods

**Key resources table**

| Reagent type or resource | Designation | Source of reference | Identifiers | Additional information |
|---|---|---|---|---|
| Gene (*C. elegans*) | *npr-1* | Worm base | Wormbase ID: WBGene00003807 | Sequence: C39E6.6 |
| Gene (*C. elegans*) | *glb-5* | Worm base | Wormbase ID: WBGene00015964 | Sequence: C18C4.1 |
| Gene (*C. elegans*) | *dpy-10* | Worm base | Wormbase ID: WBGene00001072 | Sequence: T14B4.7 |
| Gene (*C. elegans*) | *daf-22* | Worm base | Wormbase ID: WBGene00013284 | Sequence: Y57A10C.6 |
| Strain, strain background (*E. coli*) | OP50 | Caenorhabditis genetics center (CGC) | RRID:WB-STRAIN: OP50 | |
| Strain, strain background (*E. coli*) | OP50 GFP | Caenorhabditis genetics center (CGC) | RRID:WB-STRAIN: OP50-GFP | with pFPV25.1 express GFP. |
| Strain (*C. elegans*) | N2 | Cori Bargmann Lab (The Rockefeller University) | RRID:WB-STRAIN:N2 | |
| Strain (*C. elegans*) | CB4856 | *Caenorhabditis elegans* Natural Diversity Resource (CeNDR) | RRID:WB-STRAIN: CB4856 | Website: https://www.elegansvariation.org/ |
| Strain (*C. elegans*) | DL238 | *Caenorhabditis elegans* Natural Diversity Resource (CeNDR) | RRID:WB-STRAIN: DL238 | Website: https://www.elegansvariation.org/ |
| Strain (*C. elegans*) | JU258 | *Caenorhabditis elegans* Natural Diversity Resource (CeNDR) | RRID:WB-STRAIN: JU258 | Website: https://www.elegansvariation.org/ |
| Strain (*C. elegans*) | JU775 | *Caenorhabditis elegans* Natural Diversity Resource (CeNDR) | RRID:WB-STRAIN: JU775 | Website: https://www.elegansvariation.org/ |
| Strain (*C. elegans*) | MY16 | *Caenorhabditis elegans* Natural Diversity Resource (CeNDR) | RRID:WB-STRAIN: MY16 | Website: https://www.elegansvariation.org/ |
| Strain (*C. elegans*) | MY23 | *Caenorhabditis elegans* Natural Diversity Resource (CeNDR) | RRID:WB-STRAIN: MY23 | Website: https://www.elegansvariation.org/ |
| Strain (*C. elegans*) | CX11314 | *Caenorhabditis elegans* Natural Diversity Resource (CeNDR) | RRID:WB-STRAIN :CX11314 | Website: https://www.elegansvariation.org/ |
| Strain (*C. elegans*) | LKC34 | *Caenorhabditis elegans* Natural Diversity Resource (CeNDR) | RRID:WB-STRAIN: LKC34 | Website: https://www.elegansvariation.org/ |
| Strain (*C. elegans*) | ED3017 | *Caenorhabditis elegans* Natural Diversity Resource (CeNDR) | RRID:WB-STRAIN: ED3017 | Website: https://www.elegansvariation.org/ |

*Continued on next page*

*Continued*

| Reagent type or resource | Designation | Source of reference | Identifiers | Additional information |
|---|---|---|---|---|
| Strain (*C. elegans*) | JT11398 | *Caenorhabditis elegans* Natural Diversity Resource (CeNDR) | RRID:WB-STRAIN:JT11398 | Website: https://www.elegansvariation.org/ |
| Strain (*C. elegans*) | EG4725 | *Caenorhabditis elegans* Natural Diversity Resource (CeNDR) | RRID:WB-STRAIN:EG4725 | Website: https://www.elegansvariation.org/ |
| Strain (*C. elegans*) | PTM229 | This paper | RRID:WB-STRAIN:PTM229 | Strain Background: N2 |
| Strain (*C. elegans*) | PTM288 | This paper | RRID:WB-STRAIN:PTM288 | Strain Background: N2 |
| Strain (*C. elegans*) | PTM289 | This paper | RRID:WB-STRAIN:PTM289 | Strain Background: N2 |
| Strain (*C. elegans*) | PTM95 | PMID: 27467070 | RRID:WB-STRAIN:PTM95 | Strain Background: N2 |
| Strain (*C. elegans*) | CX12311 | PMID: 21849976 | RRID:WB-STRAIN:CX12311 | Strain Background: N2 |
| Strain (*C. elegans*) | QG1 | PMID: 27172189 | RRID:WB-STRAIN:QG1 | Strain Background: N2 |
| Strain (*C. elegans*) | CX10774 | PMID: 19285466 | RRID:WB-STRAIN:CX10774 | Strain Background: N2 |
| Strain (*C. elegans*) | CX11400 | PMID: 23284308 | RRID:WB-STRAIN:CX11400 | Strain Background: CB4856 |
| Strain (*C. elegans*) | CX4148 | PMID: 9741632 | RRID:WB-STRAIN:CX4148 | Strain Background: N2 |
| Strain (*C. elegans*) | DA609 | PMID: 9741632 | RRID:WB-STRAIN:DA609 | Strain Background: N2 |
| Strain (*C. elegans*) | CX7102 | PMID: 16903785 | RRID:WB-STRAIN:CX7102 | Strain Background: N2 |
| Strain (*C. elegans*) | PTM400 | This paper | RRID:WB-STRAIN:PTM400 | Strain Background: N2 |
| Strain (*C. elegans*) | PTM401 | This paper | RRID:WB-STRAIN:PTM401 | Strain Background: N2 |
| Strain (*C. elegans*) | PTM402 | This paper | RRID:WB-STRAIN:PTM402 | Strain Background: N2 |
| Strain (*C. elegans*) | PTM403 | This paper | RRID:WB-STRAIN:PTM403 | Strain Background: N2 |
| Strain (*C. elegans*) | PTM404 | This paper | RRID:WB-STRAIN:PTM404 | Strain Background: N2 |
| Strain (*C. elegans*) | PTM405 | This paper | RRID:WB-STRAIN:PTM405 | Strain Background: N2 |
| Strain (*C. elegans*) | PTM408 | This paper | RRID:WB-STRAIN:PTM408 | Strain Background: N2 |
| Sequence-based reagents (Plasmid) | Plasmid: pDD162 PrU6::*dpy-10_sgRNA* | PMID: 27467070 | | CRISPR/Cas9 gene editing sgRNA |
| Sequence-based reagents (Plasmid) | Plasmid: pDD162 P*reft3::Cas9* | PMID: 27467070 | | CRISPR/Cas9 gene editing Cas9 |
| Sequence-based reagents (Oligonucleotide) | *dpy-10 (cn64)* | PMID: 25161212 | | CRISPR/Cas9 gene editing DNA repair oligo for inducing *dpy-10 cn64* mutation |

*Continued on next page*

*Continued*

| Reagent type or resource | Designation | Source of reference | Identifiers | Additional information |
|---|---|---|---|---|
| Sequence-based reagents (Oligonucleotide) | *dpy-10 (kah82/kah83)* | This paper | | CRISPR/Cas9 gene editing DNA repair oligo for inducing *dpy-10* Thr90 slient mutation |
| Sequence-based reagents (Oligonucleotide) | *dpy-10 (kah84)* | This paper | | CRISPR/Cas9 gene editing DNA repair oligo for inducing *dpy-10* Arg92 slient mutation |
| Chemical compound, drug | 1x Antibiotic-Antimycotic | ThermoFisher | Cat. No.: 15240062 | |
| Chemical compound, drug | FUDR | Sigma | Cat. No.: F0503 | |
| Commercial assay, kit | Taqman probe: *dpy-10 (kah82/kah83)* | ThermoFisher: Custom TaqMan SNP Genotyping Assays | PTM09 | |
| Commercial assay, kit | Taqman probe: *dpy-10 (kah84)* | ThermoFisher: Custom TaqMan SNP Genotyping Assays | PTM10 | |
| Commercial assay, kit | Taqman probe: *npr-1(g320)* | ThermoFisher: Custom TaqMan SNP Genotyping Assays | PTM08 | |
| Commercial assay, kit | Taqman probe: WBVar00209467 | ThermoFisher: Custom TaqMan SNP Genotyping Assays | PTM11 | |
| Commercial assay, kit | TruSeq Stranded mRNA kit | Illumina | Cat. No.: 20020595 | |
| Commercial assay, kit | Zymo DNA isolation kit | Zymo | Cat. No.: D4071 | |
| Commercial assay, kit | Zymo DNA cleanup kit | Zymo | Cat. No.: D4064 | |
| Commercial assay, kit | ddPCR Supermix for Probes | BIORAD | Cat. No.: 1863010 | |
| Commercial assay, kit | Droplet Generation Oils | BIORAD | Cat. No.: 1863005 | |
| Commercial assay, kit | ddPCR Droplet Reader Oil | BIORAD | Cat. No.: 1863004 | |
| Commercial assay, kit | VECTASHIELD antifade Mounting Medium with DAPI | VECTOR | Cat. No.: H-1200 | |
| Software, Algorithm | edgeR | PMID: 19910308 | RRID:SCR_012802 | Opensource: https://bioconductor.org/packages/release/bioc/html/edgeR.html |
| Software, Algorithm | SARTools | PMID: 27280887 | RRID:SCR_016533 | Opensource: https://github.com/PF2-pasteur-fr/SARTools |
| Software, Algorithm | MATLAB | MathWorks | RRID:SCR_001622 | |
| Software, Algorithm | Rstudio | Rstudio | RRID:SCR_000432 | https://www.rstudio.com/ |
| Software, Algorithm | JMP12 | SAS JMP | RRID:SCR_014242 | |
| Software, Algorithm | Image J | NIH | RRID:SCR_003070 | Opensource: https://imagej.nih.gov/ij/ |

*Continued on next page*

*Continued*

| Reagent type or resource | Designation | Source of reference | Identifiers | Additional information |
|---|---|---|---|---|
| Software, Algorithm | MetaMorph | Molecular Devices | RRID:SCR_002368 | |
| Software, Algorithm | Custom TaqMan Assay Design Tool | ThermoFisher | | https://www.thermofisher.com/order/custom-genomic-products/tools/genotyping/ |

## Strains

The following strains were used in this study

### Wild strains

N2; CB4856; DL238; JU258; JU775; MY16; MY23; CX11314; LKC34; ED3017; JT11398; EG4725. The N2 strain originated from the Bargmann lab (The Rockefeller University). The remaining eleven wild strains came from the *Caenorhabditis elegans* Natural Diversity Resource (*Cook et al., 2017*).

### Barcoded strains

PTM229 *dpy-10 (kah82)II*; PTM288 *dpy-10 (kah83)II kyIR1(V, CB4856 >N2) qgIR1(X, CB4856 >N2)*; PTM289 *dpy-10 (kah84)II kyIR1(V, CB4856 >N2) qgIR1(X, CB4856 >N2)*; The barcoded strains were generated using previously published reagents for modifying the *dpy-10* gene (*Arribere et al., 2014*). Two modified repair oligos with the following sequence were used to edit silent mutations into the 90th (Thr) or 92nd amino acid (Arg):

*dpy-10* 90th silent mutation:

5'-CACTTGAACTTCAATACGGCAAGATGAGAATGACTGGAAACCGTACTGCTCGTGGTGCCTA
TGGTAGCGGAGCTTCACATGGCTTCAGACCAACAGCCTAT-3'

*dpy-10* 92nd silent mutation:

5'-CACTTGAACTTCAATACGGCAAGATGAGAATGACTGGAAACCGTACCGCTCGCGGTGCCTA
TGGTAGCGGAGCTTCACATGGCTTCAGACCAACAGCCTAT-3'

### The microinjection mix was

50 ng/uL P*eft3::Cas9*, 25 ng/uL *dpy-10* sgRNA, 500 nM *dpy-10(cn64)* repair oligo, and one of the 500 nM *dpy-10(90/92)* repair oligo. This mix was injected into N2 or CX12311 and so-called 'jackpot broods' were identified by the presence of a large number of F1 animals with the roller phenotype. From these plates, wildtype animals were singled and genotyped using Sanger-sequencing. *kah82* and *kah83* contain the 90th Thr silent mutation (ACC - > ACT). *kah84* contains the 92nd Arg silent mutation (CGT - > CGC).

### Near isogenic lines

CX12311 *kyIR1(V, CB4856 >N2) qgIR1(X, CB4856 >N2)*; QG1 *qgIR1(X, CB4856 >N2)*; CX10774 *kyIR1(V, CB4856 >N2)*; CX11400 *kyIR9(X, N2 >CB4856)*. These strains were originally described in previous studies (*Bendesky et al., 2012*; *Bernstein and Rockman, 2016*; *McGrath et al., 2009*; *McGrath et al., 2011*).

### *npr-1* loss of function

CX4148 *npr-1(ky13)X*; DA609 *npr-1(ad609)X*; these strains were previously described (*de Bono and Bargmann, 1998*).

### URX, AQR, PQR genetic ablation strains

qaIs2241[*Pgcy-35::GFP Pgcy-36::egl-1 lin15+*] is an integrated transgene that genetically ablates URX, AQR, and PQR neurons (*Chang et al., 2006*). This transgene was crossed into a number of introgressed regions using standard genetic techniques. CX7102 *qaIs2241X*; PTM400 *qaIs2241X*; PTM401 *qgIR1(X, CB4856 >N2) qaIs2241X*; PTM402 *kyIR1 (V, CB4856 >N2) qaIs2241X*; PTM403 *kyIR1(V, CB4856 >N2) qgIR1(X, CB4856 >N2) qaIs2241X*;.

### *daf-22* strains

*daf-22(kah8)II* is a *daf-22* gene disruption made by CRISPR/Cas9 genome editing (*Large et al., 2016*). The deletion of *daf-22* spans the 6th Pro to 219th Glu (The deleted sequence is: 5′-caaaggta-tacatcgttggagtcggtatgacaaagttttgtaagccggga...ggatcaggtgatcaatgcccgtaagatctacgactt-tatgggtctcctcg-3′). This transgene was crossed into a number of introgressed regions using standard genetic techniques. PTM95 *daf-22(kah8)II kyIR1(V, CB4856 >N2) qgIR1(X, CB4856 >N2)*; PTM404 *daf-22(kah8)II dpy10(kah83)II*; PTM405 *daf-22(kah8)II*; PTM408 *daf-22(kah8)II kyIR1(V, CB4856 >N2) qgIR1(X, CB4856 >N2)*.

## Growth conditions

Animals were grown following standard conditions. With exceptions listed below, animals were culti-vated on modified nematode growth medium (NGM) plates containing 2% agar seeded with 200 µl of an overnight culture of the *E. coli* strain OP50 in an incubator set at 20°C. Strains were grown for at least three generations without starvation before any assays were conducted. For assays manipu-lating the environmental $O_2$ levels, animals were grown inside a BioSpherix C474 chamber using a BioSpherix C21 single chamber controller to control ambient $O_2$ levels. For these assays, animals were not grown in temperature incubators, and the room temperature was typically kept ~21°C. For competition experiments on non-burrowing plates, 1.25% agarose and 0.75% agar replaced the agar concentrations of normal growth plates. To create uniform lawns, liquid cultures of OP50 bacte-ria were poured onto plates to cover the entire surface area of the plate and then poured off.

### UV treatment

9 cm NGM plates were seeded with 300 µL of an overnight culture of the *E. coli* strain OP50 and placed at room temperature for 2 days. Then plates were placed in Stratagene UV Stratalinker 2400 with 254 nm radiation. The lids were removed and the plates were irradiated at 9999 mJ/cm². The efficacy of the killing was measured as described previously {*Gems and Riddle (2000)* #82}.

## Pairwise fitness measurements

Competition experiments were performed as previously (*Large et al., 2016*). Briefly, Ten L4 stage animals from each strain were picked onto 9 cm NGM plates seeded with 300 µL of an overnight *E. coli* OP50 culture and incubated at room temperature for 3 days. After 5 days, animals were trans-ferred to an identically prepared NGM plate and then subsequently transferred every 4 days for five to seven generations. For transfers, animals were washed off from the test plates using M9 buffer and collected into 1.5 mL centrifuge tube. The animals were mixed by inversion and allowed to stand for approximately one minute to settle adult animals. 50 uL of the supernatant containing ~1000–2000 L1-L2 animals were seeded on next plates. The remaining animals were con-centrated and placed in a −80°C freezer for future genomic DNA isolation. Genomic DNA was col-lected from every odd generation using a Zymo DNA isolation kit (D4071).

To quantify the relative proportion of each strain, we used a digital PCR based approach using a custom TaqMan probe (Applied Biosciences). Genomic DNA was digested with EcoRI for 30 min at 37°C. The digested products were purified using a Zymo DNA cleanup kit (D4064) and diluted to ~1 ng/µL for the following Taqman assay. Four TaqMan probes were designed using ABI custom soft-ware that targeted the *dpy-10 (kah82)*, *dpy-10 (kah84)*, *npr-1(g320)*, or SNP WBVar00209467 in *glb-5*. These probes were validated using defined concentrations of DNA from animals containing each allele. The Taqman digital PCR assays were performed using a Biorad QX200 digital PCR machine with standard probe absolute quantification protocol. The relative allele proportion was calculated for each DNA sample using count number of the droplet with fluorescence signal (*Equation 1*). To calculate the relative fitness of the two strains using three to four measurements of relative fitness, we used linear regression to fit this data to a one-locus generic selection model (*Equations 2 and 3*), assuming one generation per transfer.

$$P(A)_t = \frac{No.\ Allele\ A}{No.\ Allele\ A + No.\ Allele\ a} \tag{1}$$

$$P(A)_t = \frac{P(A)_0 W_{AA}^t}{P(A)_0 W_{AA}^t + \left(1 - P(A)_0\right) W_{aa}^t} \tag{2}$$

$$log\left(\frac{\frac{P(A)_0}{P(A)_t} - P(A)_0}{1 - P(A)_0}\right) = \left(log\left(\frac{W_{aa}}{W_{AA}}\right)\right) t \tag{3}$$

## Aerotaxis assays

To measure bordering rates, 2-week-old NGM plates were removed from a 4°C cold room, seeded with 200 µL of *E. coli* OP50 and incubated for 2 days at room temperature. 150 adult animals were picked onto these assay plates and placed in either a 20°C incubator or a BioSpherix chamber for 3 hr. Bordering behavior was quantified using a dissecting microscope by identifying animals whose whole body resided within 1 mm of the border of the bacteria lawn.

## Transcriptome analysis

N2 and CX12311 L4 hermaphrodites were picked to fresh agar plates. Their adult progeny were synchronized using alkaline-bleach to isolate eggs. These eggs were washed three times using M9 buffer and placed on a tube roller overnight to allow eggs to hatch. About 400 L1 animals were placed on NGM agar plates seeded with non-uniform lawns of *E. coli* OP50 and incubated in a BioSpherix chamber set at 10% $O_2$ or 21% $O_2$ levels for 48 hr. The ~L4 stage animals were washed off and used for standard Trizol RNA isolation. Replicates were performed on different days. The RNA libraries for next-generation sequencing were prepared using an Illumina TruSeq Stranded mRNA kit (20020595) following its standard protocol. These libraries were sequenced using an Illumina NextSeq 500 platform. Reads were aligned using HISAT2 using default parameters for pair-end sequencing. Transcript abundance was calculated using HTseq and then used as inputs for the SARTools (*Varet et al., 2016*). Within this R package, edgeR is used for normalization and differential analysis. N2 cultured at 21% $O_2$ is treated as wild type (*Chen et al., 2014*). The genes showing significantly different expression ($log_2$(fold) >1 or $log_2$(fold) < −1, FDR adjusted p-value<0.01) were selected to perform Hierarchical Cluster analysis, and Principal Component analysis. Sequencing reads were uploaded to the SRA under PRJNA437304.

## Food consumption assays

### Liquid food consumption

The 96 well-plates were prepared by pipetting 150 µL S media containing *E. coli* OP50 with density OD600 of 1.0 (CFU ~ $0.8 \times 10^9$/mL), 500 uM FUDR and 1 x Antibiotic-Antimycotic (ThermoFisher 15240062). 20 synchronized animals (L4 stage or young adult) were put into each well, pipetting to mix each well completely before using BioTek Synergy H4 multimode plate reader to record OD600 optical density every 24 hr from Day one to Day 5.

### Plate food consumption

The 24-well plates were prepared by pipetting 0.75mL NGM agar contain 25 µM FUDR and 1x Antibiotic-Antimycotic (ThermoFisher 15240062) to each well. The freshly prepared plates were placed in fume hood and dried with air flow for 1.5 hr. 20 µL of freshly cultured OD600 of 4.0 (CFU ~ $3.2 \times 910^9$/mL) *E. coli* OP50-GFP(pFPV25.1) were seeded in the center of each well. Animals were synchronized using alkaline-bleach. The eggs were washed by M9 buffer for three times and rotating on tube roller overnight to allow eggs to hatch. About 200 L1 animals were placed on NGM agar plates seeded with *E. coli* OP50 and cultivate at 20° C or BioSpherix chamber at 21° C for 50 hr. Ten animals (Late L4 stage or young adult) were transferred to each well of the first five columns of the food consumption assay 24 well-plates. The remaining four wells were used as control wells to measure the GFP signal degradation. After placing animals on the food consumption assay plates, the fluorescence signal of OP50-GFP from each well was quantified by area scanning protocol using BioTek Synergy H4 multimode plate reader at 6 mm height as the starting time point. The 24-well plates were then incubated in incubator or BioSpherix chamber for 18 hr and the fluorescence signal were quantified again as the ending time point. The fluorescence signal at end time point from each

well was normalized using the fluorescence signal degradation amount of control wells. The normalization was performed using the equation as below:

$$Fluorescence\_Control(0hr) = \beta.Fluorescence\_Control(18hrs) \tag{1}$$

All the signals from control wells were used to do linear regression and estimate coefficient β. The estimated amounts of bacteria at 18 hr for each test is:

$$Fluorescence(18hrs\_normalized) = \beta Fluorescence(18hrs) \tag{2}$$

The food consumption for each well was calculated by:

$$Food\,consumption\,amount = Fluorescence(0hr) - Fluorescence(18\,hr\_normalized). \tag{3}$$

## Pharyngeal pumping and size assays

Animals were synchronized using alkaline-bleach. The eggs were washed by M9 buffer for three times and rotating on tube roller overnight to allow eggs to hatch. About 200 L1 animals were placed on NGM agar plates seeded with *E. coli* OP50 and cultivated at 20°C for 72 hr. In the pharyngeal pumping rates assays, the pharynges of 10 young adult animals (72 hr after place L1 on NGM agar plate) were observed for 30 s each in three separate trails. To measure the pharyngeal size, young adult animals were placed onto agar pad and immobilized by 25 mM NaN$_3$. For each strain, pharyngeal sizes of 30 animals from three different plates were imaged under 40x objective lens using z-stack DIC microscope. The diameter of pharyngeal metacorpus, diameter of terminal bulb diameter, procorpus length, and isthmus length were measured using ImageJ software.

## Reproductive timing and growth assays

To measure reproductive timing, animals were synchronized by picking 10 adult animals onto an NGM plate, allowing them to lay eggs for two hours, and then removing the adult animals from the plate. These offspring were then monitored using a 12x dissecting microscope at indicated time points to count the number of animals with oocytes and fertilized eggs in their uterus. A subset of these animals was washed off at indicated time points and fixed in 95% ethanol. The nuclei were stained with 1.5 µg/mL DAPI solution in Vectashield antifade mounting medium (VECTOR H-1200) for 10 min in the dark before visualization. Each spermatheca was imaged by z-stack fluorescence microscopy using a 100x lens to determine whether spermatogenesis had started or to count the number of sperm produced by the hermaphrodite.

Reproductive rate and body size measurements were measured as described previously (*Large et al., 2016*).

## Modifications for pheromone assays

For crude pheromone assays, crude pheromone was prepared as described previously (*Zhuo et al., 2017*). The crude pheromone was resuspended in ethanol and stored in −20°C. A dauer formation assay was performed to test the efficacy of crude pheromone. A 1/333 (v/v) crude pheromone level could induce a high >80% rate of dauer formation in N2 animals grown on 20 uL of heat killed *E. coli* OP50 bacteria (5 mg/mL).

For the feeding assay and reproductive timing assays, the animals were grown on NGM plates for three generations on plates containing the indicated concentrations of crude pheromone (or ethanol control).+indicates a 1/10,000 (v/v) crude pheromone and ++indicated a 1/2000 (v/v) ratio of crude pheromones. The plates were then dried in biosafety cabinet for 1.5 hr, seeded with 200 µL of overnight culture of the *E. coli* strain OP50, incubated overnight, and used immediately for experiments.

## Exploration assay

35 mm Petri dishes evenly seeded with OP50 *E. coli* Bacteria for 24 hr before the start of assay. Individual L4 hermaphrodites were placed in the center of the plate and cultivated in BioSpherix chamber in 10% O$_2$ or 21% O$_2$ level at 21° C for 3 hr. The plates were placed on a grid that has 105 squares which cover the whole plate. The number of full or partial squares that contained animal's tracks was quantified and the exploration fraction was calculated (*Equation 1*).

$$\text{Exploration fraction} = \frac{\text{No. grids contained tracks}}{105} \qquad (1)$$

## Statistics

All raw data are included in figure source data tables. All replicates were biological replicates using animals grown independently for multiple generations. The number of biological replicates were chosen using power analysis based upon the standard deviation from previous assays. To assess statistical significance, we performed one-way ANOVA tests followed by Tukey's honest significant difference test to correct for multiple comparisons or the Wilcoxon-Mann-Whitney nonparametric test for pairwise comparisons. The Friedman test was used to compare the reproductive timing assays. The exact test used is listed in the legend for each panel.

## Video files

These files show a single generation (3 days) of growth of the N2 or CX12311 grown strain in the presence of 21% or 10% environmental $O_2$.

## Acknowledgements

Some strains were provided by the Andersen lab, the Bargmann lab, and the CGC, which is funded by NIH Office of Research Infrastructure Programs (P40 OD010440). We thank Dalia Gulick in Greg Gibson's laboratory for assistance with the Bio Rad QX200 instrument used for digital PCR experiments performed in the Georgia Tech Petit-IBB Genomics Core. We thank Weipeng Zhuo for use of crude pheromones and Biao Zeng for suggestions regarding gene expression analysis. We are also grateful to Erik Andersen, Nicole Baran, Cori Bargmann, Levi Morran, Annalise Paaby, Will Ratcliff, and members of the McGrath lab for comments on the manuscript.

## Additional information

### Funding

| Funder | Grant reference number | Author |
| --- | --- | --- |
| National Institute of General Medical Sciences | R01GM114170 | Yuehui Zhao<br>Lijiang Long<br>Wen Xu<br>Richard F Campbell<br>Edward E Large<br>Patrick T McGrath |
| National Institute on Aging | R21AG050304 | Patrick T McGrath |
| Ellison Medical Foundation | New Scholars in Aging | Lijiang Long<br>Wen Xu<br>Patrick T McGrath |

The funders had no role in study design, data collection and interpretation, or the decision to submit the work for publication.

### Author contributions

Yuehui Zhao, Conceptualization, Data curation, Formal analysis, Validation, Investigation, Visualization, Methodology, Writing—original draft, Writing—review and editing; Lijiang Long, Data curation, Formal analysis, Validation, Visualization, Methodology, Writing—review and editing; Wen Xu, Richard F Campbell, Data curation, Formal analysis, Validation, Methodology, Writing—review and editing; Edward E Large, Resources, Writing—review and editing; Joshua S Greene, Methodology, Writing—review and editing; Patrick T McGrath, Conceptualization, Formal analysis, Supervision, Funding acquisition, Visualization, Methodology, Writing—original draft, Project administration, Writing—review and editing

## Author ORCIDs

Yuehui Zhao http://orcid.org/0000-0002-9496-0023
Wen Xu http://orcid.org/0000-0003-2085-7223
Patrick T McGrath http://orcid.org/0000-0002-1598-3746

## Decision letter and Author response

Decision letter https://doi.org/10.7554/eLife.38675.059
Author response https://doi.org/10.7554/eLife.38675.060

## Additional files

### Supplementary files

• Supplementary file 1. Full list of average read count per gene of each strain in 10% O2 or 21% O2 level. The table contains the full list of average read count per gene. Each gene count number per strain/condition is the average of 6 replicates.
DOI: https://doi.org/10.7554/eLife.38675.054

• Transparent reporting form
DOI: https://doi.org/10.7554/eLife.38675.055

### Data availability

Sequencing data have been deposited in SRA under PRJNA437304. Source data file have been provided for all figures.

The following dataset was generated:

| Author(s) | Year | Dataset title | Dataset URL | Database and Identifier |
|---|---|---|---|---|
| Zhao Y, Long L, McGrath PT | 2018 | Quantify fitness and dissect pleiotropic traits to study adaptation strategy of laboratory domesticated C. elegans strain N2 | https://www.ncbi.nlm.nih.gov/bioproject/PRJNA437304/ | NCBI BioProject, PRJNA437304 |

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
