## [Decision Letter]

Thank you for submitting your article "Solitary feeding behavior in *Caenorhabditis elegans* is a nonadaptive side effect of extended laboratory evolution" for consideration by *eLife*. Your article has been reviewed by three peer reviewers, including Yuichi Iino as the Reviewing Editor and Reviewer #1, and the evaluation has been overseen by Patricia Wittkopp as the Senior Editor. The following individual involved in review of your submission has agreed to reveal their identity: Young-Jai You (Reviewer #2).

The reviewers have discussed the reviews with one another and the Reviewing Editor has drafted this decision to help you prepare a revised submission.

Summary:

In their manuscript, McGrath and colleagues explore the reason for N2 fitness under laboratory conditions. Or in other words, what is the selective advantage of the *glb-5*(Bristol)/*npr-1*(215V) allelic combination. Using competition assays, NGS, behavioral, and physiological measurements they show that the above allelic combination increases feeding efficiency and fertility. Moreover, they show that this fitness advantage is regulated by the URX-RMG circuit in a pheromone-dependent manner.

The necessary experiments were performed very nicely and thoroughly. The logic for each experiment is clearly written. The data are presented very nicely.

Essential revisions:

1) Because *daf-22* is required for peroxisomal fatty acid β-oxidation which is important for metabolism and growth, it is not clear whether the *daf-22* effects in fitness is solely due to absence of pheromone. In Figure 5C and D, authors observed a strong effect of the *daf-22* mutation on both number of eggs and food intake. Reviewers strongly suggest that authors try adding crude pheromones to the *daf-22* mutants to see whether these defects are (at least partially) rescued by addition of pheromones. Reviewers are aware that if the results are not as expected (rescue of the defects), it does not necessarily exclude the possibility that lack of the pheromones is the cause, but if it does rescue, the results will strengthen the authors' claim. If authors decide not to perform these tests, they need to explicitly state the possibility of the metabolic effect described above.

2) Previous studies show that eating live OP50 decreases worms' lifespan (Garigan et al., 2002; Gems and Riddle, 2000). Could it be that the derived *glb-5/npr-1* make N2 more resistant to OP50-toxicity? If so, the fitness of CX12311 should be similar to N2 on dead OP50. Authors need to test this possibility.

3) All reviewers agree that in the current manuscript, it is unclear how 'increased feeding rate', 'changes to the neural circuits' and 'controlling pheromone responses' are connected. This is partly due to the writing, namely scarcity (or lack) of discussion on the mechanical effect of pheromones. Authors just say, for example, "changes to social/solitary behavior in N2 were a pleiotropic consequence of changes to integrated O2 and pheromone neural circuits that regulate feeding rate" (Abstract), but no further statements. Based on the results provided, regulation of the URX-RMG circuit or BAG, ADF or ASG by pheromone is likely to be controlling the fitness. In fact, ADL/ASK/RMG neurons have been shown to respond to pheromone in an npr-1-dependent manner in Macosko et al., 2009 (Barmann lab) and in https://doi.org/10.1073/pnas.1621274114 (de Bono lab), for example. Also, a recent study by the de Bono lab shows that "N2 animals were more strongly repelled by 10 nM C9 than *npr-1* animals" (https://doi.org/10.1073/pnas.1618934114). Moreover, they show that the O2-sensor *gcy-35* is important for the attraction of *npr-1* worms to pheromones. Authors need to cite these papers and discuss the possibility that this circuit may be regulated by pheromones and in turn regulate food intake. They do not need to do further experiments to test the hypotheses, but discussing it (as above or in some other way) will improve the readability of the paper.

4) Figure 5A suggests that the derived allele of *glb-5*, i.e. *glb-5*(Bri), decreases the fitness of worms by acting in other neurons than AQR, PQR, and URX (e.g. BAG, ADF, or ASG). However, this is not proven, because, as indicated by the authors, *glb-5* introgressed region is ~290 kb in size, and therefore the observed results can be due to variations other than that of *glb-5*. If the authors are to claim that *glb-5*(Bri) acts in neurons other than AQR, PQR, URX, they need to generate QG1 transgenes expressing *glb-5*(Haw) (the non-derived *glb-5*) in these neurons or BAG, ADF, ASG (or non-neural tissues, Persson et al., 2009) and compare their fitness and/or food-consumption.

5) The title is overselling the results. Specifically, the word "side effect" is inappropriate and should be rephrased.

---

## [Author Response]

Essential revisions:1) Because daf-22 is required for peroxisomal fatty acid β-oxidation which is important for metabolism and growth, it is not clear whether the daf-22 effects in fitness is solely due to absence of pheromone. In Figure 5C and D, authors observed a strong effect of the daf-22 mutation on both number of eggs and food intake. Reviewers strongly suggest that authors try adding crude pheromones to the daf-22 mutants to see whether these defects are (at least partially) rescued by addition of pheromones. Reviewers are aware that if the results are not as expected (rescue of the defects), it does not necessarily exclude the possibility that lack of the pheromones is the cause, but if it does rescue, the results will strengthen the authors' claim. If authors decide not to perform these tests, they need to explicitly state the possibility of the metabolic effect described above.

In our original submission, we primarily discussed the *daf-22* phenotype in the context of its role in producing pheromones. As the reviewers rightly pointed out, *daf-22* also has a role in peroxisomal fatty acid β-oxidation. As requested, we performed experiments to attempt to rescue the fitness-proximal traits using partially purified crude pheromone extracts. We focused on food consumption and reproductive timing assays, using two concentrations of crude pheromone. In these experiments, we were unable to rescue the *daf-22* phenotype for either trait.

There are a number of potential reasons that prevent us from ruling out a role for pheromones in these traits – 1) the crude pheromone purification is taken from supernatant produced from worms grown in liquid culture which likely alters the release/concentrations of many of the individual components, 2) the purification procedure using lyophilization followed by ethanol extraction might not purify all the individual pheromone components equally, 3) the concentrations we have used are not physiological for these traits, and 4) pheromones might be necessary but not sufficient, and additional *daf-22*-dependent molecular pathways also contribute.

It is beyond the scope of work that can be accomplished in the two month deadline to try to distinguish between the various roles that *daf-22* can play. For this resubmission, we have chosen to include all new data as figures, and have rewritten the text to downplay the role that pheromones play in the fitness circuits throughout the paper. We have also included a Discussion paragraph to discuss the possible roles that *daf-22* might play to modify fitness and food consumption rates, including in regulating metabolism.

2) Previous studies show that eating live OP50 decreases worms' lifespan (Garigan et al., 2002; Gems and Riddle, 2000). Could it be that the derived glb-5/npr-1 make N2 more resistant to OP50-toxicity? If so, the fitness of CX12311 should be similar to N2 on dead OP50. Authors need to test this possibility.

We have performed competition experiments between N2 and CX12311 on heat-killed OP50 and a mock-killed OP50 bacteria control. In both conditions, the N2 strain was more fit than the CX12311 animals, indicating resistance to OP50 toxicity cannot be the only cause of fitness differences between the strains. There was a significant quantitative difference in relative fitness between the two conditions, however, we point out that directly comparing these values is difficult, as the heat-killed bacteria do not support the same level of population growth as a live bacteria plate. We have included these data in Figure 3.

3) All reviewers agree that in the current manuscript, it is unclear how 'increased feeding rate', 'changes to the neural circuits' and 'controlling pheromone responses' are connected. This is partly due to the writing, namely scarcity (or lack) of discussion on the mechanical effect of pheromones. Authors just say, for example, "changes to social/solitary behavior in N2 were a pleiotropic consequence of changes to integrated O2 and pheromone neural circuits that regulate feeding rate" (Abstract), but no further statements. Based on the results provided, regulation of the URX-RMG circuit or BAG, ADF or ASG by pheromone is likely to be controlling the fitness. In fact, ADL/ASK/RMG neurons have been shown to respond to pheromone in an npr-1-dependent manner in Macosko et al., 2009 (Barmann lab) and in https://doi.org/10.1073/pnas.1621274114 (de Bono lab), for example. Also, a recent study by the de Bono lab shows that "N2 animals were more strongly repelled by 10 nM C9 than npr-1 animals" (https://doi.org/10.1073/pnas.1618934114). Moreover, they show that the O2-sensor gcy-35 is important for the attraction of npr-1 worms to pheromones. Authors need to cite these papers and discuss the possibility that this circuit may be regulated by pheromones and in turn regulate food intake. They do not need to do further experiments to test the hypotheses, but discussing it (as above or in some other way) will improve the readability of the paper.

We have updated the Discussion to discuss more directly the neural circuit mechanisms that could lead to changes in food consumption. However, our inability to rescue *daf-22* with crude pheromone has led us to focus less on the potential role that pheromone-circuits play and instead focus more on the role URX might play in regulating feeding rate. We also discuss more broadly how *daf-22* could regulate fitness and feeding either through its role in creating pheromones or through additional pathways it participates in.

4) Figure 5A suggests that the derived allele of glb-5, i.e. glb-5(Bri), decreases the fitness of worms by acting in other neurons than AQR, PQR, and URX (e.g. BAG, ADF, or ASG). However, this is not proven, because, as indicated by the authors, glb-5 introgressed region is ~290 kb in size, and therefore the observed results can be due to variations other than that of glb-5. If the authors are to claim that glb-5(Bri) acts in neurons other than AQR, PQR, URX, they need to generate QG1 transgenes expressing glb-5(Haw) (the non-derived glb-5) in these neurons or BAG, ADF, ASG (or non-neural tissues, Persson et al., 2009) and compare their fitness and/or food-consumption.

After discussion with the reviewing editor regarding the feasibility of these experiments in the timeframe provided, we have rephrased this statement to remove this speculation.

*5) The title is overselling the results. Specifically, the word "side effect" is inappropriate and* should be rephrased.

We have rewritten the title.